# Deep Reinforcement Learning for Cost-Effective Medical Diagnosis

**Zheng Yu**[*]
Princeton University

**Yikuan Li**[*]
Northwestern University

**Joseph C. Kim**[*]
Princeton University

**Kaixuan Huang**[*]
Princeton University

**Yuan Luo**[†]
Northwestern University

**Mengdi Wang**[†]
Princeton University

## Abstract

Dynamic diagnosis is desirable when medical tests are costly or time-consuming. In this work, we use reinforcement learning (RL) to find a dynamic policy that selects lab test panels sequentially based on previous observations, ensuring accurate testing at a low cost. Clinical diagnostic data are often highly imbalanced; therefore, we aim to maximize the F1 score instead of the error rate. However, optimizing the non-concave $F_1$ score is not a classic RL problem, thus invalidating standard RL methods. To remedy this issue, we develop a reward shaping approach, leveraging properties of the $F_1$ score and duality of policy optimization, to provably find the set of all Pareto-optimal policies for budget-constrained $F_1$ score maximization. To handle the combinatorially complex state space, we propose a Semi-Model-based Deep Diagnosis Policy Optimization (SM-DDPO) framework that is compatible with end-to-end training and online learning. SM-DDPO is tested on diverse clinical tasks: ferritin abnormality detection, sepsis mortality prediction, and acute kidney injury diagnosis. Experiments with real-world data validate that SM-DDPO trains efficiently and identify all Pareto-front solutions. Across all tasks, SM-DDPO is able to achieve state-of-the-art diagnosis accuracy (in some cases higher than conventional methods) with up to $85\%$ reduction in testing cost. Core codes are available on GitHub[1].

## 1 Introduction

In clinical practice, physicians usually order multiple panels of lab tests on patients and their interpretations depend on medical knowledge and clinical experience. Each test panel is associated with certain financial cost. For lab tests within the same panel, automated instruments will simultaneously provide all tests, and eliminating a single lab test without eliminating the entire panel may only lead to a small reduction in laboratory cost (Huck & Lewandrowski, 2014). On the other hand, concurrent lab tests have been shown to exhibit significant correlation with each other, which can be utilized to estimate unmeasured test results (Luo et al., 2016). Thus, utilizing the information redundancy among lab tests can be a promising way of optimizing which test panel to order when balancing comprehensiveness and cost-effectiveness. The efficacy of the lab test panel optimization can be evaluated by assessing the predictive power of optimized test panels on supporting diagnosis and predicting patient outcomes.

We investigate the use of reinforcement learning (RL) for lab test panel optimization. Our goal is to dynamically prescribe test panels based on available observations, in order to maximize diagnosis/prediction accuracy while keeping testing at a low cost. It is quite natural that sequential test panel selection for prediction/classification can be modeled as a Markov decision process (MDP).

---

[*]Co-first authors.

[†]Co-senior authors.

Contact information: Z. Yu, J. Kim, K. Huang and M. Wang:{zhengy, josephck, kaixuanh, mengdiw}@princeton.edu; Y. Li and Y. Luo:{yikuan.li, yuan.luo}@northwestern.edu

[1]https://github.com/Zheng321/Deep-Reinforcement-Learning-for-Cost-Effective-Medical-Diagnosis

However, application of reinforcement learning (RL) to this problem is nontrivial for practical considerations. One practical challenge is that clinical diagnostic data are often highly imbalanced, in some cases with <5% positive cases (Khushi et al., 2021; Li et al., 2010; Rahman & Davis, 2013). In supervised learning, this problem is typically addressed by optimizing towards accuracy metrics suitable for unbalanced data. The most prominent metric used by clinicians is the F1 score, i.e., the harmonic mean of a prediction model's recall and precision, which balances type I and type II errors in a single metric. However, the F1 score is not a simple weighted error rate - this makes designing the reward function hard for RL. Another challenge is that, for cost-sensitive diagnostics, one hopes to view this as a multi-objective optimization problem and fully characterize the cost-accuracy tradeoff, rather than finding an ad-hoc solution on the tradeoff curve. In this work, we aim to provide a tractable algorithmic framework, which provably identifies the set of all Pareto-front policies and trains efficiently. Our main contributions are summarized as follows:

- We formulate cost-sensitive diagnostics as a multi-objective policy optimization problem. The goal is to find all optimal policies on the Pareto front of the cost-accuracy tradeoff.
- To handle severely imbalanced clinical data, we focus on maximizing the $F_1$ score directly. Note that $F_1$ score is a nonlinear, nonconvex function of true positive and true negative rates. *It cannot be formulated as a simple sum of cumulative rewards, thus invalidating standard RL solutions.* We leverage monotonicity and hidden minimax duality of the optimization problem, showing that the Pareto set can be achieved via a reward shaping approach.
- We propose a Semi-Model-based Deep Diagnostic Policy Optimization (SM-DDPO) method for learning the Pareto solution set from clinical data. Its architecture comprises three modules and can be trained efficiently by combing pretraining, policy update, and model-based RL.
- We apply our approach to real-world clinical datasets. Experiments show that our approach exhibits good accuracy-cost trade-off on all tasks compared with baselines. Across the experiments, our method achieves state-of-the-art accuracy with up to $80\%$ reduction in cost. Further, SM-DDPO is able to compute the set of optimal policies corresponding to the entire Pareto front. We also demonstrate that SM-DDPO applies not only to the $F_1$ score but also to alternatives such as the AM score.

## 2 RELATED WORK

Reinforcement learning (RL) has been applied in multiple clinical care settings to learn optimal treatment strategies for sepsis Komorowski et al. (2018), to customize antiepilepsy drugs for seizure control Guez et al. (2008) etc. See survey Yu et al. (2021) for more comprehensive summary. Guidelines on using RL for optimizing treatments in healthcare has also been proposed around the topics of variable availability, sample size for policy evaluation, and how to ensure learned policy works prospectively as intended Gottesman et al. (2019). However, using RL for simultaneously reducing the healthcare cost and improving patient's outcomes has been underexplored.

Our problem of cost-sensitive dynamic diagnosis/prediction is closely related to feature selection in supervised learning. The original static feature selection methods, where there exists a common subset of features selected for all inputs, were extensively discussed Guyon & Elisseeff (2003); Kohavi & John (1997); Bi et al. (2003); Weston et al. (2003; 2000). Dynamic feature selection methods He et al. (2012); Contardo et al. (2016); Karayev et al. (2013), were then proposed to take the difference between inputs into account. Different subsets of features are selected with respect to different inputs. By defining certain information value of the features Fahy & Yang (2019); Bilgic & Getoor (2007), or estimating the gain of acquiring a new feature would yield Chai et al. (2004). Reinforcement learning based approaches Ji & Carin (2007); Trapeznikov & Saligrama (2013); Janisch et al. (2019); Yin et al. (2020); Li & Oliva (2021); Nam et al. (2021) are also proposed to dynamically select features for prediction/classification. We give a more detailed discussion in Appendix A.

## 3 PARETO-FRONT PROBLEM FORMULATION

### 3.1 MARKOV DECISION PROCESS (MDP) MODEL

We model the dynamic diagnosis/prediction process for a new patient as an episodic Markov decision process (MDP) $\mathcal{M} = (\mathcal{S}, \mathcal{A}, P, R, \gamma, \xi)$. As illustrated in Figure 1, the state of a patient is described by $s = \mathbf{x} \odot M$, where $\mathbf{x} \in \mathbb{R}^d$ denotes $d$ medical tests of a patient, $M \in \{0, 1\}^d$ is a binary mask

indicating whether the entries of $\mathbf{x}$ are observed or missing. Let there be $D$ test panels, whose union is the set of all $d$ tests. The action set $\mathcal{A} = \{1, 2, \cdots, D\} \sqcup \{P, N\}$ contains two sets of actions – observation actions and prediction/diagnosis actions. At each stage, one can either pick an action $a \in \{1, 2, \cdots, D\}$ from any one of the available panels, indicating choosing a test panel $a$ to observe, which will incur a corresponding observation cost $c(a)$; Or one can terminate the episode by directly picking a prediction action $a \in \{P, N\}$, indicating diagnosing the patient as positive class (P) or negative class (N). A penalty will generated if the diagnosis does not match the ground truth $y$. An example of this process in sepsis mortality prediction is illustrated in Figure 1. We considers the initial distribution $\xi$ to be patients with only demographics panel observed and discount factor $\gamma = 1$.

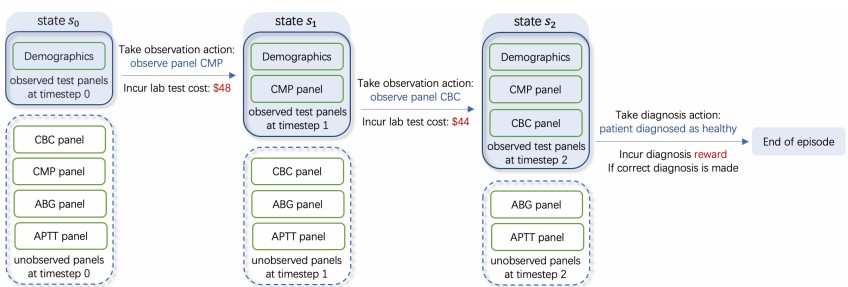

Figure 1: MDP model of dynamic diagnosis: illustration of state-action transitions in one episode.

### 3.2 MULTI-OBJECTIVE POLICY OPTIMIZATION FORMULATION

Let $\pi : \mathcal{S} \to \mathcal{A}$ be the overall policy, a map from the set of states to the set of actions. We optimize $\pi$ towards two objectives:

• **Maximizing prediction accuracy.** Due to severely imbalanced data, we choose to maximize the $F_1$ score[2], denoted by $F_1(\pi)$, as a function of policy $\pi$. $F_1$ score measures the performance of the diagnosis by considering both type I and type II errors, which is defined as:

$$F_1(\pi) = \frac{\text{TP}(\pi)}{\text{TP}(\pi) + \frac{1}{2}(\text{FP}(\pi) + \text{FN}(\pi))} = \frac{2\text{TP}(\pi)}{1 + \text{TP}(\pi) - \text{TN}(\pi)}$$

where $\text{TP}(\pi), \text{TN}(\pi), \text{FP}(\pi), \text{FN}(\pi)$ are normalized true positive, true negative, false positive and false negative that sum up to 1. Remark that $\text{TP}(\pi), \text{TN}(\pi), \text{FP}(\pi), \text{FN}(\pi)$ can all be expressed as sum of rewards/costs over the MDP's state trajectories. *However, $F_1(\pi)$ is nonlinear with respect to the MDP's state-action occupancy measure, thus it cannot be expressed as any cumulative sum of rewards.*

• **Lowering cost**. Define the testing cost by $\text{Cost}(\pi) = \mathbb{E}^\pi[\sum_{t \geq 0} \sum_{k \in [D]} c(k) \cdot \mathbf{1}\{a_t = k\}]$, where $\mathbb{E}^\pi$ denotes expectation under policy $\pi$, $c(k)$ is the cost of panel $k$.
In this work, we hope to solve for cost-sensitive policies for all possible testing budget. In other words, we aim to find the cost-sensitive Pareto front as follows.

**Definition 3.1** (**Cost-$F_1$ Pareto Front of Multi-Objective Policy Optimization**). *The Pareto front $\Pi^*$ for cost-sensitive dynamic diagnosis/prediction is the set of policies such that*

$$\Pi^* = \cup_{B>0} \underset{\pi}{\text{argmax}}\{F_1(\pi) \text{ subject to } \text{Cost}(\pi) \leq B\} \tag{1}$$

Finding $\Pi^*$ requires novel solutions beyond standard RL methods. Challenges are two-folded: **(1)** Even in the single-objective case, $F_1(\pi)$ is a nonlinear, non-concave function of $\text{TP}(\pi), \text{TN}(\pi)$. Although both $\text{TP}(\pi), \text{TN}(\pi)$ can be formulated as expected sum of rewards in the MDP, the $F_1$ score is never a simple sum of rewards. Standard RL methods do not apply to maximizing such a function. **(2)** We care about finding the set of all Pareto-optimal policies when there are two conflicting objectives, rather than an ad hoc point on the trade-off curve.

---

[2]An alternative to the F1 score is the AM metric that measures the average of true positive rate and true negative rate for imbalanced data Natarajan et al. (2018); Menon et al. (2013). Our approach directly applies to such linear metric. Please refer to Appendix F for details.

## 4 Finding Cost-$F_1$ Pareto Front via Reward Shaping

The $F_1$ score is a nonlinear and nonconvex function of true positive, true negative, false positive, false negative rates. It cannot be expressed by sum of rewards. This invalidates all existing RL methods even in the unconstrained case, creating tremendous challenges.

Despite the non-concavity and nonlinearity of $F_1$, we will leverage the mathematical nature of Markov decision process and properties of the F1 score to solve problem (1). In this section, we provide an optimization duality analysis and show how to find solutions to problem (1) via reward shaping and solving a reshaped cumulative-reward MDP.

**Step 1: utilizing monotonicity of $F_1$ score** To start with, we note that $F_1$ score is monotonically increasing in both TP and TN. Assume, for any given cost budget $B$, the optimal policy $\pi^*(B)$ achieves the highest $F_1$ score. Then $\pi^*(B)$ is also optimal to the following program:

$$\max_{\pi} \left\{ \text{TN}(\pi) \text{ subject to Cost}(\pi) \leq B, \text{TP}(\pi) \geq \text{TP}(\pi^*(B)) \right\},$$

indicating the Pareto front of $F_1$ score is a subset of

$$\Pi^* \subseteq \cup_{B>0, K\in[0,1]} \underset{\pi}{\text{argmax}} \left\{ \text{TN}(\pi) \text{ subject to Cost}(\pi) \leq B, \text{TP}(\pi) \geq K \right\}. \tag{2}$$

**Step 2: reformulation using occupancy measures** Fix any specific pair $(B, B')$. Consider the equivalent dual linear program form Zhang et al. (2020) of the above policy optimization problem (2). It is in terms of the cumulative state-action occupancy measure $\mu : \Delta_{\mathcal{A}}^{\mathcal{S}} \to \mathbb{R}_{\geq 0}^{\mathcal{S} \times \mathcal{A}}$, defined as:

$\mu^{\pi}(s, a) := \mathbb{E}^{\pi} \left[ \sum_{t \geq 0} \mathbf{1}(s_t = s, a_t = a) \right], \forall s \in \mathcal{S}, a \in \mathcal{A}$. Then the program (2) is equivalent to:

$$\max_{\mu} \text{TN}(\mu) \text{ subject to Cost}(\mu) \leq B, \text{ TP}(\mu) \geq K, \sum_{a} \mu(s,a) = \sum_{s',a' \in [D]} \mu(s', a') P(s|s', a') + \xi(s), \forall s$$

where $\xi(\cdot)$ denotes initial distribution, and TP, TN and cost are reloaded in terms of occupancy $\mu$ as:

$$\text{TP}(\mu) = \sum_{y=\text{P}, a=\text{P}} \mu(s,a), \text{ TN}(\mu) = \sum_{y=\text{N}, a=\text{N}} \mu(s,a), \text{ Cost}(\mu) = \sum_{k \in [D]} c(k) \cdot \sum_{s,a=k} \mu(s,a).$$

**Step 3: utilizing hidden minimax duality** The above program can be equivalently reformulated as a max-min program:

$$\max_{\mu} \min_{\lambda \geq 0, \rho \leq 0} \text{TN}(\mu) + \lambda \cdot (\text{TP}(\mu) - K) + \rho \cdot (\text{Cost}(\mu) - B)$$

$$\text{subject to } \sum_{a} \mu(s,a) = \sum_{s',a' \in [D]} \mu(s', a') P(s|s', a') + \xi(s), \forall s.$$

Note the max-min objective is linear in terms of $\lambda, \rho$ and $\mu$. Thus, minimax duality holds, then we can swap the min and max to obtain the equivalent form:

$$\min_{\lambda \geq 0, \rho \leq 0} \max_{\mu} \text{TN}(\mu) + \lambda \cdot (\text{TP}(\mu) - K) + \rho \cdot (\text{Cost}(\mu) - B)$$

$$\text{subject to } \sum_{a} \mu(s,a) = \sum_{s',a' \in [D]} \mu(s', a') P(s|s', a') + \xi(s), \forall s.$$

For any fixed pair of $(\lambda, \rho)$, the inner maximization problem of the above can be rewritten equivalently into an unconstrained policy optimization problem: $\max_{\pi} \text{TN}(\pi) + \lambda \cdot \text{TP}(\pi) + \rho \cdot \text{Cost}(\pi)$. This is finally a standard cumulative-sum MDP problem, with reshaped reward: reward $\rho \cdot c(t)$ for the action of choosing test panel $t$, reward $\lambda$ for the diagnosis action and get a true positive, reward 1 for getting a true negative. Putting together three steps, we can show the following theorem. The full proof can be found in Appendix E.

**Theorem 4.1.** *The Cost-$F_1$ Pareto front defined in* (1) *is a subset of the collection of all reward-shaped solutions, given by*

$$\Pi^* \subseteq \overline{\Pi} := \cup_{\lambda \geq 0, \rho \leq 0} \underset{\pi}{\text{argmax}} \left\{ TN(\pi) + \lambda \cdot TP(\pi) + \rho \cdot Cost(\pi) \right\}.$$

Thus, to learn the full Pareto front, it suffices to solve a collection of unconstrained policy optimization problems with reshaped cumulative rewards.

## 5 METHOD

In this section, we propose a deep reinforcement learning pipeline for Pareto-optimal dynamic diagnosis policies. We use a modular architecture for efficient encoding of partially-observed patient information, policy optimization and reward learning.

### 5.1 ARCHITECTURE

Our Semi-Model-based Deep Diagnostic Policy Optimization (SM-DDPO) framework is illustrated in Figure 2. The complete dynamic testing policy $\pi$ comprises three models: (1) a posterior state encoder for mapping partially-observed patient information to an embedding vector; (2) a state-to-diagnosis/prediction classifier which can be reviewed as a reward function approximator; (3) a test panel selector that outputs an action based on the encoded state. This modular architecture makes RL tractable via a combination of pre-training, policy update and model-based RL.

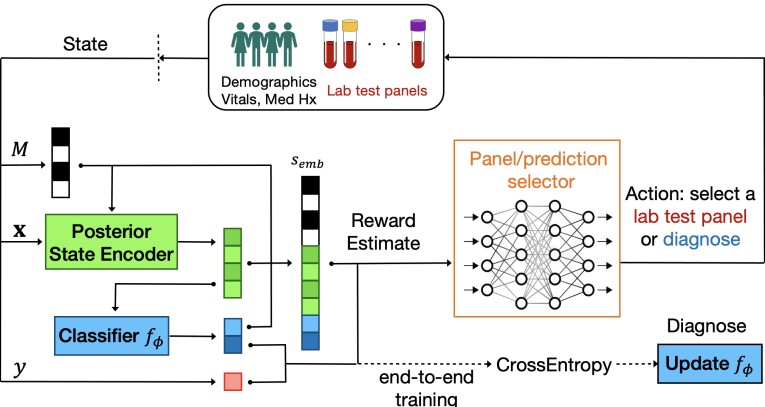

Figure 2: Dynamic diagnostic policy learning via semi-model-based proximal policy optimization. The full policy $\pi$ comprises of three modules: posterior state encoder, classifier, and panel selector.

### 5.2 POSTERIOR STATE ENCODER

We borrowed the idea of imputation to map the partially observed patient information to a posterior embedding vector. In this work, we consider a flow-based deep imputer named *EMFlow*[3]. Given the imputer $\mathrm{Imp}_\theta(\cdot)$ parameterized by $\theta$, the RL agent observes tests $\mathbf{x}\odot M$ and calculates $\mathrm{Imp}_\theta(\mathbf{x}\odot M) \in \mathbb{R}^d$ as a posterior state encoder. Unlike conventional imputation Lin & Tsai (2020); Austin et al. (2021); Osman et al. (2018), our posterior state encoder aims at resolving exponentially many possible missing patterns. Therefore, we pretrain it on unlabeled augmented data, constructed by repeatedly and randomly masking entries as additional samples.

### 5.3 END-TO-END TRAINING VIA SEMI-MODEL-BASED POLICY UPDATE

Training the overall policy using a standard RL algorithm alone (such as Q learning, or policy gradient) would suffer from the complex state and action spaces. To ease the heavy training, we design a semi-model-based *modular* approach to train the panel selector and classifier concurrently but in different manners:

• The classifier $f_\phi(\cdot) : \mathbb{R}^d \to \mathbb{R}^2$, parameterized by $\phi$, maps the posterior encoded state $\mathrm{Imp}_\theta(\mathbf{x}\odot M)$ to a probability distribution over labels. It is trained by directly minimizing the cross entropy loss $\ell_c$ from collected data[4]. This differs from typical classification in that the data are collected adaptively by RL, rather than sampled from a prefixed source.

• The panel selector, a network module parameterized by $\psi$, takes the following as input

$$s^{\mathrm{emb}}_{\theta,\phi}(s) = s^{\mathrm{emb}}_{\theta,\phi}(\mathbf{x}\odot M) = (\mathrm{Imp}_\theta(\mathbf{x}\odot M), f_\phi(\mathrm{Imp}_\theta(\mathbf{x}\odot M)), M), \qquad (3)$$

---

[3]The *EMFlow imputation method is originally proposed in Ma & Ghosh (2021) that maps the data space to a Gaussian latent space via normalizing flows. We give a more detailed discussion of this method in Appendix C.*

[4]The forms of the training objective $\ell_c$ and $\ell_{rl}$ of classifier and panel selector are given in Appendix D

and maps it to a probability distribution over actions. We train the panel selector by using the classical proximal policy updates (PPO) Schulman et al. (2017), which aims at maximizing a clipped surrogate objective regularized by a square-error loss of the value functions and an entropy bonus. We denote this loss function as $\ell_{rl}$ and relegate its expanded form to Appendix D.

• The full algorithm updates panel selector and classifier concurrently, given in Algorithm 1 and visualized in Figure 2. We call it "semi-model-based" because it maintains a running estimate of the classifier (which is a part of the reward model of the MDP) while making proximal policy update.

---

**Algorithm 1** Semi-Model-Based Deep Diagnosis Policy Optimization (SM-DDPO)

---

**Initialize:** $\text{Imp}_\theta$, Classifier $f_{\phi^{(0,0)}}$, Panel/Prediction Selection Policy $\pi_{\psi^{(0,0)}}$, number of loops $L, L_1, L_2$, stepsize $\eta$;
**for** $i = 0, 1, \cdots, L$ **do**                                                                    ▷ End-to-end training outer loop
    Construct RL environment using state embedding $s_{\theta,\phi^{(i,0)}}^{\text{emb}}$ defined in (3)
    **for** $j = 1, 2, \cdots, L_1$ **do**                                                      ▷ Policy update inner loop
        Run RL policy in environment for $T$ timesteps and save observations in $Q$
        Update panel selection policy by $\psi^{(i,j)} = \text{argmax}_\psi \ell_{rl}(\psi; \psi^{(i,j-1)})$
    **end for**
    Set $\psi^{(i+1,0)} = \psi^{(i,L_1)}$
    **for** $j = 1, 2, \cdots, L_2$ **do**                                                      ▷ Classifier update inner loop
        Sample minibatch $B_j$ from $Q$
        Update classifier by $\phi^{i,j} = \phi^{i,j-1} - \eta \cdot \nabla_\phi \frac{1}{|B_j|} \sum_{k \in B_j} \ell_c(\phi^{i-1}; (\mathbf{x_k}, M_k))$
    **end for**
    Set $\phi^{(i+1,0)} = \phi^{(i,L_2)}$
**end for**
**Output:** Classifier $f_{\phi^{(L+1,0)}}$, Policy $\pi_{\psi^{(L+1,0)}}$.

---

Such a hybrid RL technique of model learning and policy updates has been used for solving complex games, where a notable example is Deepmind's Muzero Schrittwieser et al. (2020). Further, we remark that Algorithm 1 does end-to-end training. Thus, it is compatible with on-the-fly learning. The algorithm can start with as little as zero knowledge about the prediction task, and it can keep improving on new incoming patients by querying test panels and finetuning the state encoder.

## 6 EXPERIMENTS

We test the method on three clinical tasks using real-world datasets. See Table 1 for a summary. We split each dataset into 3 parts: training set, validation set, and test set. Training data is further split into two disjoint sets, one for pretraining the state encoder and the other one for end-to-end RL training. Validation set is used for tuning hyperparameters [5]. During RL training [6], we sample a random patient and a random subset of test results as initial observations at the beginning of an episode, for sufficient exploration. We evaluate the trained RL policy on patients from the test sets, initialized at a state with zero observed test result, and report F1 score and AUROC.

Table 1: Summary statistics of ferritin, AKI and sepsis datasets

| Dataset | # of tests | # test panels | # patients | % positive class | #training | #validation | #held-out testing |
|---------|-----------|---------------|------------|-------------------|-----------|-------------|-------------------|
| Ferritin | 39 | 6 | 43,472 | 8.9% | 32,602 | 6,522 | 4,248 |
| AKI | 19 | 4 | 23,950 | 16.5% | 17,964 | 3,600 | 2,386 |
| Sepsis | 28 | 4 | 5,783 | 14.5% | 4,335 | 869 | 579 |

### 6.1 CLINICAL TASKS

We briefly describe three clinical tasks for our experiments. We refer to Appendix B for more details.

**Ferritin abnormality detection** Blood ferritin level can indicate abnormal iron storage, which is commonly used to diagnose iron deficiency anemia Short & Domagalski (2013) or hemochromatosis

---

[5]Detailed data splitting, hyperparameter choices and searching ranges are presented in Appendix B.
[6]Our codes used the implementation of PPO algorithm in package Raffin et al. (2021).

(iron overload) Crownover & Covey (2013). Machine learning models can predict abnormal ferritin levels using concurrent laboratory measurements routinely collected in primary care Luo et al. (2016); Kurstjens et al. (2022). These predictive models achieve promising results, e.g. around 0.90 AUC using complete blood count and C-reactive protein Kurstjens et al. (2022), and around 0.91 AUC using common lab tests Luo et al. (2016). However, both studies required the full observation of all selected predictors without taking the financial costs into consideration.

We applied our proposed models to a ferritin dataset from a tertiary care hospital (approved by Institutional Review Board), following the steps described in Luo et al. (2016). Our dataset includes 43,472 patients, of whom 8.9% had ferritin levels below the reference range that should be considered abnormal. We expected to predict abnormal ferritin results using concurrent lab testing results and demographic information. These lab tests were ordered through and can be dynamically selected from 6 lab test panels, including basic metabolic panel (BMP, n=9, [estimated national average] cost=$36), comprehensive metabolic panel (CMP, n=16, cost=$48), basic blood count (BBC, n=10, cost=$26), complete blood count (CBC, n=20, cost=$44), transferrin saturation (TSAT, n=2, cost=$40) and Vitamin B-12 test (n=1, cost=$66).

**Acute Kidney Injury Prediction** Acute kidney injury (AKI) is commonly encountered in adults in the intensive care unit (ICU), and patients with AKI are at risk for adverse clinical outcomes such as prolonged ICU stays and hospitalization, need for renal replacement therapy, and increased mortality Kellum & Lameire (2013). AKI usually occurs over the course of a few hours to days and the efficacy of intervention greatly relies on the early identification of deterioration Kellum & Lameire (2013). Prior risk prediction models for AKI based on EHR data yielded modest performance, e.g., around 0.75 AUC using a limited set of biomarkers Perazella (2015) or a specific group of patients Sanchez-Pinto & Khemani (2016), or around 0.8 AUC using comprehensive lab panels of general adult ICU populations such as from the MIMIC datasetZimmerman et al. (2019); Sun et al. (2019).

In this experiment, we followed steps in Zimmerman et al. (2019) to extract 23,950 ICU visits of 19,811 patients from the MIMIC-III dataset Johnson et al. (2016), among which 16.5% patients develop AKI during their ICU stay. We aimed at predicting the AKI onset within 72 hours of ICU admission using a total of 31 features including demographics, physiologic measurements, and lab testing results extracted within 24 hours of ICU admission. The lab tests were categorized into 4 panels, i.e. CBC (n=3, cost=$44), CMP (n=8, cost=$48), the arterial blood gas panel (ABG, n=2, cost=$473) and the activated partial thromboplastin time panel (APTT, n=6, cost=$26). The demographic information and physiologic measurements were collected before test panel selection, thus they are considered visible.

**Sepsis Mortality Prediction for ICU Patients** Sepsis is a life-threatening organ dysfunction, and is a leading cause of death and cost overruns in ICU patients Angus & Van der Poll (2013) Early identification of risk of mortality in septic patients is important to evaluate the patients' status and improve their clinical outcomes Moreno et al. (2008). Most of the previous sepsis mortality prediction models use all available test results as predictors, without balancing the cost of ordering all the associated test panels Lee et al. (2020); Ding & Luo (2021); Shin et al. (2021); Moreno et al. (2008).

We followed steps in Shin et al. (2021) to collect 5,783 septic patients from the MIMIC-III dataset Johnson et al. (2016) according to the Sepsis-3 criteria Singer et al. (2016). The in-hospital mortality rate of this cohort is 14.5%. We focused on predicting in-hospital mortality for these sepsis patients using demographics information, medical histories, mechanical ventilation status, the first present lab testing results and physiologic measurements within 24 hours of ICU admission, and the Sequential Organ Failure Assessment (SOFA) score. Similarly to the setup in the AKI experiment, the lab tests were also categorized into 4 panels of CBC (n=5), CMBP (n=15), ABG (n=6) and APTT (n=2). The components of SOFA score may be based on the lab testing results in CBC, CMP or ABG panels Singer et al. (2016). Demographic features are considered visible.

## 6.2 Performance Results

Our method is tested with comparison to a number of baselines that use either full/partial test results or statically/dynamically selected tests for prediction. They include logistic regression, random forest Ho (1995), XGBoost Chen & Guestrin (2016), LightGBM Ke et al. (2017), a 3-layer multi-layer perceptron, as well as RL-based approach Janisch et al. (2019). Experiments results, such as F1 score, AUROC and testing costs are reported in Table 2. We emphasize that these baselines are incapable to handle the task of finding the Pareto front. Thus, we only test them under no budget constraints.

Table 2: Model performance, measured by $F_1$ score, area under ROC (AUC), and testing cost, for three real-world clinical datasets. The tested models include logistic regression (LR), random forests (RF), gradient boosted regression trees (XGBoost Chen & Guestrin (2016) and LightGBM Ke et al. (2017) ), 3-layer multi-layer perceptron, Q-learning for classification with costly features (CWCF)Janisch et al. (2019), Random Selection (RS), Fixed Selection (FS). All models were fine-tuned to maximize the $F_1$ score. The model yielded the highest $F_1$ score is in bold. The model required the least testing cost is underlined. More detailed results of this table with more dynamic baselines and standard deviations reported in Appendix B.

| Models | Ferritin | | | AKI | | | Sepsis | | | Test Selection |
|---|---|---|---|---|---|---|---|---|---|---|
| *Metrics* | $F_1$ | *AUC* | *Cost* | $F_1$ | *AUC* | *Cost* | $F_1$ | *AUC* | *Cost* | Strategy |
| LR | 0.539 | 0.935 | $290 | 0.452 | 0.797 | $591 | 0.506 | 0.825 | $591 | Full |
| RF | 0.605 | 0.938 | $290 | 0.439 | 0.764 | $591 | 0.456 | 0.801 | $591 | Full |
| XGBoost | 0.617 | 0.938 | $290 | 0.404 | 0.785 | $591 | 0.431 | 0.828 | $591 | Full |
| LightGBM | **0.627** | **0.941** | **$290** | 0.474 | 0.790 | $591 | 0.500 | 0.844 | $591 | Full |
| 3-layer DNN | 0.616 | 0.938 | $290 | 0.494 | 0.802 | $591 | 0.517 | 0.845 | $591 | Full |
| LR (2 panels) | 0.401 | 0.859 | $92 | 0.473 | 0.797 | $92 | 0.488 | 0.811 | $92 | Fixed |
| RF (2 panels) | 0.504 | 0.887 | $92 | 0.425 | 0.768 | $92 | 0.478 | 0.828 | $92 | Fixed |
| XGBoost (2 panels) | 0.519 | 0.895 | $92 | 0.410 | 0.781 | $92 | 0.459 | 0.877 | $92 | Fixed |
| LightGBM (2 panels) | 0.571 | 0.901 | $92 | 0.491 | 0.792 | $92 | 0.502 | 0.864 | $92 | Fixed |
| FS | 0.585 | 0.927 | $74 | 0.434 | 0.787 | $98 | 0.500 | 0.837 | $90 | Fixed |
| RS | 0.437 | 0.845 | $145 | 0.424 | 0.748 | $295 | 0.473 | 0.789 | $295 | Random |
| CWCF | 0.554 | 0.718 | $256 | 0.283 | 0.510 | $326 | 0.112 | 0.503 | $301 | Dynamic |
| SM-DDPO$_{pretrained}$ | 0.607 | 0.925 | $80 | **0.519** | **0.789** | **$90** | **0.567** | **0.836** | **$85** | Dynamic |
| SM-DDPO$_{end2end}$ | 0.624 | 0.928 | $62 | 0.495 | 0.795 | $97 | 0.562 | 0.845 | $90 | Dynamic |

• **Comparisons with baseline models using full observation of data.** The results are presented in Table 2. Across all three clinical tasks, our proposed model can achieve comparable or even state-of-the-art performance, while significantly reducing the financial cost. On sepsis dataset, SM-DDPO$_{end2end}$ yielded better results ($F_1$=0.562, AUROC=0.845) than the strongest baseline models, LightGBM ($F_1$=0.517, AUROC=0.845), when saving up to 84% in test cost. On ferritin dataset, LightGBM ($F_1$=0.627, AUROC=0.948) performed slightly better than our model ($F_1$=0.624, AUROC=0.928), however, by using 5x testing cost. On AKI dataset, SM-DDPO$_{end2end}$ ($F_1$=0.495, AUROC=0.795) achieved comparable results to the optimal full observation model, 3-layer MLP ($F_1$=0.494, AUROC=0.802), while saving the testing cost from $591 to $90.

• **Comparisons with other test selection strategies.** Our proposed SM-DDPO$_{end2end}$, using RL-inspired dynamic selection strategy, consistently yielded better performance and required less testing cost across all three datasets, when compared to the models using fixed or random selection strategy. For fixed test selection strategy, we first tested the classification methods using the two most relevant panels: CBC and CMP. These baselines with reduced testing cost still behaved much worse than our approach in both $F_1$ score and AUROC. We also tested another fixed selection (FS) baseline, where we chose to always observe 2 most selected test panels reported in our approach for all patients, while keeping other modules the same. Our approach outperformed FS on both $F_1$ score and AUROC while having a similar testing cost. The random selection (RS) baseline selected test panels uniformly at random and had a worse performance. Q-learning for classification with costly features (CWCF) Janisch et al. (2019) performed poorly on all three clinical datasets. We believe this is because the model uses the same network for selecting tests and learning rewards. For such imbalanced datasets, this may make the training unstable and difficult to optimize.

• **Efficiency and accuracy of end-to-end training.** The classifier and panel selector of SM-DDPO$_{end2end}$ are **both trained from the scratch**, using Algorithm 1. As in Table 2, this end-to-end training scheme gives comparable accuracy with policies that rely on a heavily pretrained classifier (SM-DDPO$_{pretrain}$). If a brand-new diagnostic/predictive task is given, our algorithm can be trained without prior data/knowledge about the new disease. It can adaptively prescribe lab tests to learn the disease model and test selection policy in an online fashion. End-to-end training is also more data-efficient and runtime efficient.

• **Interpretability.** Our algorithm is able to select test panels that are clinically relevant. For ferritin prediction, our algorithm identifies TSAT as a most important panel, which is indeed useful for detecting iron deficiency. For AKI prediction, our algorithm recommends serum creatinine level test as an important predictor for 95% of subjects, i.e., current and past serum creatinine is indicative of future AKI, expanding its utility as a biomarker de Geus et al. (2012).

### 6.3 TRAINING CURVES

We present the training curves on AKI dataset in Figure 3. We refer more results to Appendix B.

• **SM-DDPO learns the disease model.** In end-to-end training, the diagnostic classifier is trained from the scratch. It maps any partially-observed patient state to a diagnosis/prediction. We evaluate this classifier on static data distributions, in order to eliminate the effect of dynamic test selection and focus on classification quality. Figure 3 shows that the classifier learns to make the high-quality prediction with improved quality during RL, via training only on data selected by the RL algorithm.

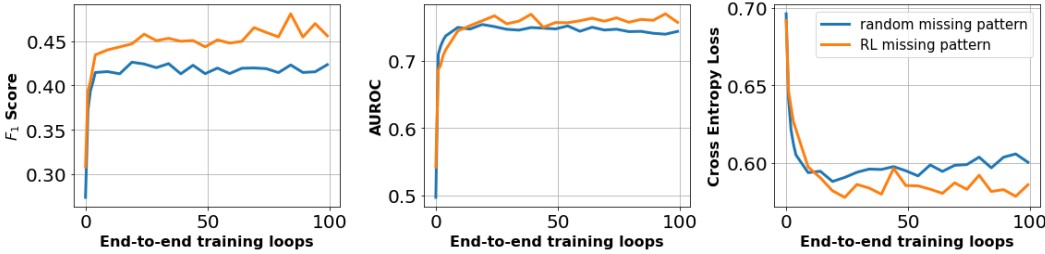

Figure 3: Classifier improvement during RL training on AKI Dataset. Accuracy of the learned classifier is evaluated on static patient distributions, with 1) random missing pattern, where we uniformly at random augment the test data; 2) missing pattern of the optimal policy's state distribution. During the end-to-end RL training, the classifier gradually improves and has higher accuracy with the second missing pattern.

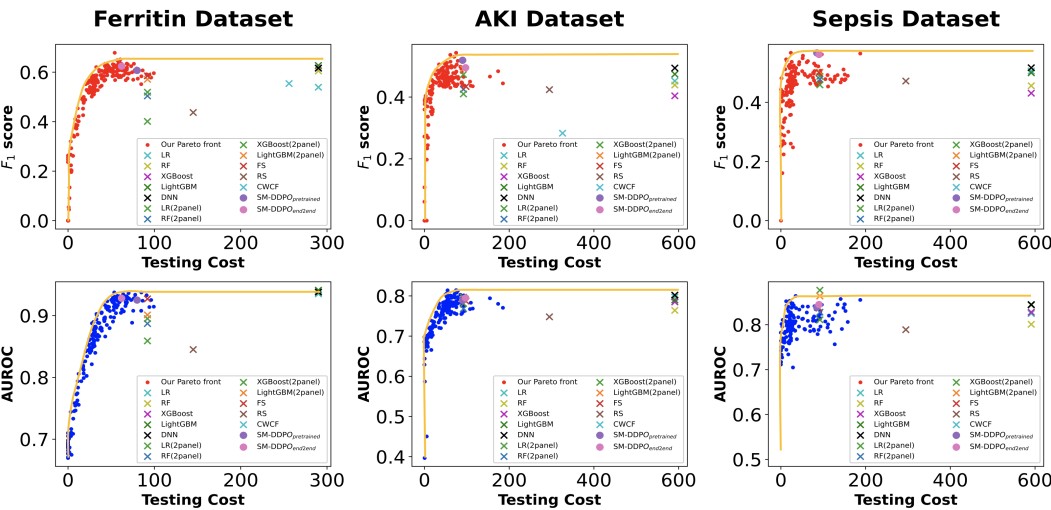

Figure 4: Cost-$F_1$ Pareto Front for maximizing $F_1$-score on Ferritin, AKI and Sepsis Datasets

### 6.4 COST-$F_1$ PARETO FRONT

Figure 4 illustrates the Pareto fronts learned on all three datasets. We trained for optimal policies on 190 MDP instances specified by different value pairs of $(\lambda, \rho)$ in Theorem 4.1, and present the corresponding performance on $F_1$ score (red) and AUROC (blue) evaluated on the test sets. We identify the Pareto front as the upper envelope of these solutions, which are the yellow curves in Figure 4. These results present the full tradeoff between testing cost and diagnostic/predictive accuracy. As a corollary, given any cost budget $B$, one is able to obtain the best testing strategy with the optimal $F_1$ performance directly from Figure 4. We present a zoom-in version in Appendix B.

## 7 SUMMARY

In this work, we develop a Semi-Model-based Deep Diagnosis Policy Optimization (SM-DDPO) method to find optimal cost-sensitive dynamic policies and achieve state-of-art performances on real-world clinical datasets with up to $85\%$ reduction in testing cost.

ACKNOWLEDGMENTS

Mengdi Wang acknowledges the support by NSF grants DMS-1953686, IIS-2107304, CMMI-1653435, ONR grant 1006977, and http://C3.AI.

Yuan Luo acknowledges the support by NIH grants U01TR003528 and R01LM013337.

Yikuan Li acknowledges the support by AHA grant 23PRE1010660.

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

## APPENDIX

We organized the appendices as follows. We first provide a complete literature review in appendix A. In appendix B, we provided the descriptive statistics of all three clinical datasets, followed by the experimental results and detailed training settings. In appendix C, we described the EMFlow imputation algorithm. In appendix D, we gave more details of the proposed end-to-end training approach. In appendix E, we proved the Theorem 4.1. In appendix F, we discuss the alternative AM metric for evaluating the diagnostic performance as an example to show the capability of our framework to handle classic linear metrics.

## A    RELATED WORK

Reinforcement learning (RL) has been applied in multiple clinical care settings to learn optimal treatment strategies for sepsis Komorowski et al. (2018), to customize antiepilepsy drugs for seizure control Guez et al. (2008) etc. See survey Yu et al. (2021) for more comprehensive summary. Guidelines on using RL for optimizing treatments in healthcare has also been proposed around the topics of variable availability, sample size for policy evaluation, and how to ensure learned policy works prospectively as intended Gottesman et al. (2019). However, using RL for simultaneously reducing the healthcare cost and improving patient's outcomes has been underexplored.

Our problem of cost-sensitive dynamic diagnosis/prediction is closely related to feature selection in supervised learning. The original static feature selection methods, where there exists a common subset of features selected for all inputs, were extensively discussed Guyon & Elisseeff (2003), including filter models (e.g., variable ranking), wrapper approaches Kohavi & John (1997), as well as integration of the feature selection in the learning process by using $\ell 1$-norm and $\ell 0$-norm Bi et al. (2003); Weston et al. (2003; 2000). Dynamic feature selection methods He et al. (2012), also named as adaptive feature acquisition Contardo et al. (2016) or feature selection with budget Karayev et al. (2013), were then proposed to take the difference between inputs into account. Different subsets of features are selected with respect to different inputs. By defining certain information value of the features Fahy & Yang (2019); Bilgic & Getoor (2007), or estimating the gain of acquiring a new feature would yield Chai et al. (2004), these methods are able to select appropriate features in an adaptive way to balance the costs of the selected features and the learning quality.

Recently, reinforcement learning based approaches Ji & Carin (2007); Trapeznikov & Saligrama (2013); Janisch et al. (2019); Yin et al. (2020); Li & Oliva (2021); Nam et al. (2021) are also proposed to dynamically select features for prediction/classification. *All these related works are directly given, or pre-define a sum of rewards as objectives.* While our task of deriving the cost-$F_1$ Pareto front, or maximizing $F_1$ score under cost budget constraints, is much harder and cannot be expressed using pre-defined reward functions. Work Nam et al. (2021); Ji & Carin (2007); Yin et al. (2020) models the feature selection process as a partially observed Markov Decision Process (POMDP). Ji & Carin (2007) points out that POMDP solutions are hard to train and cannot handle continuous-valued observations. A similar approach is proposed by Janisch et al. (2019) using a deep Q learning method. In our approach, we use a modular network and a combination of various training schemes for more efficient RL. Trapeznikov & Saligrama (2013) ranks features in a prefixed order and only requires the agent to decide if it will add a next feature or stop. The prefixed ranked order makes the problem easier to solve, but it is suboptimal compared to fully dynamic policies.

In this work, we deal with datasets with highly imbalancement, i.e., there are much fewer ill patients compared to healthy patients. To handle such data imbalancement, we use the metric of F1 score, which is a well-established and acknowledged evaluation metric in the medical domain Perazella (2015); Sanchez-Pinto & Khemani (2016), to evaluate the diagnostic performance. Alternative metrics includes linear AM metrics Natarajan et al. (2018); Menon et al. (2013) for imbalanced datasets can also handled by our frameworks, which we discussed in Appendix F.

## B    EXPERIMENTS

### B.1    DESCRIPTION OF DATASETS

We summarize a detailed description of all lab tests in three datasets in Table 3, 4 and 5

Table 3: Summary descriptive statistics of the Ferritin dataset. (N = 43472. Binary variables are reported with positive numbers and percentages. Continuous variables are reported with means and interquartile ranges.

| Variables | N / mean | % / IQR | Variables | N / mean | % / IQR |
|---|---|---|---|---|---|
| **Demographics** | | | **Complete Blood Count Panel ($44)** | | |
| Age | 50.41 | (37.00, 63.00) | [†]Abs. Nucleated RBC Count (K/$\mu$L) | 0.21 | (0.00, 0.00) |
| Race/Ethnicity | | | Abs. Basophils (K/$\mu$L) | 0.05 | (0.00, 0.10) |
| Asian | 26887 | 5.18% | Abs. Bosinophils (K/$\mu$L) | 0.16 | (0.10, 0.20) |
| Black | 6354 | 14.61% | Abs. Lymphocytes (K/$\mu$L) | 2.05 | (1.45, 2.30) |
| Hispanic | 4272 | 9.83% | Absolute Monocytes (K/$\mu$L) | 0.53 | (0.40, 0.60) |
| White | 3708 | 61.85% | Absolute Neutrophils (K/$\mu$L) | 4.13 | (2.90, 4.96) |
| Other | 2251 | 8.53% | Percent Eosinophils (%) | 2.46 | (1.00, 3.00) |
| **Comprehensive Metabolic Panel ($48)** | | | Percent Lymphocytes (%) | 29.49 | (23.67, 35.00) |
| Alanine Transaminase (UL) | 20.14 | (12.00, 21.50) | Percent Monocytes (%) | 8 | (6.00, 9.00) |
| Albumin (g/dL) | 4.3 | (4.10, 4.50) | Percent Neutrophils (%) | 60.23 | (53.50, 67.00) |
| Alkaline Phosphatase (U/L) | 67.03 | (48.00, 77.00) | Percent Basophils (%) | 0.68 | (0.00, 1.00) |
| [‡]Anion Gap (mEq/L) | 9.79 | (8.00, 12.00) | [†]White Blood Cell Count (K/$\mu$L) | 6.92 | (5.30, 7.90) |
| Aspartate Transaminase (U/L) | 22.06 | (16.00, 23.00) | [†]Red Blood Cell Count (M/$\mu$L) | 4.34 | (4.07, 4.67) |
| [‡]Bicarbonate (mmol/L) | 27.09 | (26.00, 29.00) | [†]Platelets (K/$\mu$L) | 274.78 | (224.00, 315.50) |
| [‡]Blood Urea Nitrogen (mg/dL) | 15.24 | (11.00, 17.00) | [†]Mean cell volume (fL) | 28.91 | (27.70, 30.70) |
| [‡]Calcium (mg/dL) | 9.49 | (9.20, 9.75) | [†]Mean cell hemoglobin (pg/RBC) | 32.07 | (31.38, 32.95) |
| [‡]Chloride (mmol/L) | 103.71 | (102.00, 105.00) | Mean cell hemoglobin concentration (g/dL)[†] | 90.03 | (87.00, 94.00) |
| [‡]Creatinine (mmol/L) | 0.87 | (0.69, 0.89) | [†]RBC distribution width (%) | 13.89 | (12.50, 14.60) |
| [‡]Sodium (mmol/L) | 139.27 | (138.00, 141.00) | [†]Hematocrit (%) | 38.87 | (36.60, 41.80) |
| Total Bilirubin (mg/dL) | 0.55 | (0.40, 0.60) | [†]Hemoglobin (g/dL) | 12.48 | (11.60, 13.60) |
| Total Protein (g/dL) | 7.22 | (6.90, 7.50) | **Transferrin Saturation Test ($40)** | | |
| [‡]Potassium (mmol/L) | 4.11 | (3.90, 4.30) | Iron ($\mu$g/dL) | 82.33 | (54.00, 104.00) |
| [‡]Glucose (mg/L) | 99.88 | (86.00, 102.00) | Total Iron-binding Capacity ($\mu$g/dL) | 366.81 | (321.00, 409.00) |
| Globulin (g/dL) | 3.19 | (2.70, 3.60) | **Outcome** | | |
| **Vitamin B-12 Test ($66)** | | | Abnormal Ferritin Level | 4649 | 10.69% |
| Cobalamin (pg/mL) | 547.8 | (302.00, 668.00) | | | |

†: These variables can also be ordered through basic blood count panel ($26).
‡: These variables can also be ordered through basic metabolic panel ($36).

Table 4: Summary descriptive statistics of the AKI dataset. (N = 23950. Binary variables are reported with positive numbers and percentages. Continuous variables are reported with means and interquartile ranges.)

| Variables | N/Mean | %/ IQR | Variables | N/Mean | %/ IQR |
|---|---|---|---|---|---|
| **Demographics** | | | **Comprehensive Metabolic Panel ($48)** | | |
| Gender | | | Glucose Level Maximum (mg/dL) | 173.36 | (128.00, 196.00) |
| Female | 9755 | 40.73% | Bicarbonate Level Minimum (mg/dL) | 23.87 | (21.00, 26.00) |
| Male | 14195 | 59.27% | Creatinine Level Minimum (mg/dL) | 0.74 | (0.60, 0.90) |
| Age (yr) | 60.89 | (50.66, 73.50) | Creatinine Level Maximum (mg/dL) | 0.8 | (0.60, 1.00) |
| Race/Ethnicity | | | Blood Urea Nitrogen Level Maximum (mg/dL) | 16.87 | (11.00, 20.00) |
| Black | 1688 | 6.96% | Calcium Level Minimum (mg/dL) | 8.08 | (7.62, 8.54) |
| Hispanic | 845 | 3.53% | Estimated Glomerular Filtration Rate (eGFR) | 110.2 | (80.94, 124.30) |
| White | 17261 | 72.07% | Potassium Level Maximum (mg/dL) | 4.39 | (3.90, 4.70) |
| Other | 4176 | 17.44% | **Activated Partial Thromboplastin Time Test Panel ($26)** | | |
| **Vital Signs or Bedside Measurements** | | | Partial Thromboplastin Time Maximum (s) | 32.55 | (26.60, 34.60) |
| Heart Rate Maximum (bpm) | 105.4 | (91.00, 117.00) | Partial Thromboplastin Time Minimum (s) | 40.22 | (27.80, 41.92) |
| Heart Rate Mean (bpm) | 86.97 | (76.58, 96.76) | International Normalized Ratio Minimum | 1.34 | (1.10, 1.40) |
| Systolic BP Minimum (mmHg) | 92.51 | (82.00, 102.00) | International Normalized Ratio Maximum | 1.48 | (1.20, 1.52) |
| Systolic BP Mean (mmHg) | 118.87 | (107.34, 128.45) | Prothrombin Time Minimum (s) | 14.66 | (13.10, 15.08) |
| Diastolic BP Minimum (mmHg) | 45.31 | (39.00, 52.00) | Prothrombin Time Maximum (s) | 15.63 | (13.40, 16.18) |
| Diastolic BP Mean (mmHg) | 61.98 | (54.88, 67.87) | **Complete Blood Count Panel ($44)** | | |
| Mechanical Ventilation | 12273 | 51.24% | Hemoglobin Level Minimum (g/dL) | 10.33 | (8.90, 11.70) |
| Average Urine Output (ml) | 2202.68 | (1315.00, 2792.00) | Platelet Count K/$\mu$L | 210.24 | (134.00, 261.00) |
| Temperature Maximum (celsius) | 37.6 | (37.06, 38.06) | White Blood Cell Count Maximum (mg/dL) | 12.78 | (8.50, 15.50) |
| **Arterial Blood Gas Test ($473)** | | | **Outcome** | | |
| SpO2 Minimum (%) | 92.23 | (91.00, 95.00) | Acute Kidney Injury | 3945 | 16.47% |
| SpO2 Maximum (%) | 97.41 | (96.36, 98.78) | | | |

## B.2 SUPPLEMENT EXPERIMENT RESULTS

In this section, we show a more detailed results of Table 2 presented in the main text.

## B.3 DETAILED TRAINING SETTINGS

**Computing Resource** The experiments are conducted in part through the computational resources and staff contributions provided for the Quest high performance computing facility at Northwestern University which is jointly supported by the Office of the Provost, the Office for Research, and Northwestern University Information Technology. Quest provides computing access of over 11,800 CPU cores. In our experiment, we deploy each model training job to one CPU, so that multiple

Table 5: Summary descriptive statistics of the Sepsis dataset. (N = 5396. Binary variables are reported with positive numbers and percentages. Continuous variables are reported with means and interquartile ranges.)

| Variables | N/Mean | %/ IQR | Variables | N/Mean | %/ IQR |
|---|---|---|---|---|---|
| **Demographics** | | | **Vital Signs, Bedside Measurements, or Medical Histories** | | |
| Gender | | | Heart Rate (bpm) | 90.99 | (77.00, 104.00) |
|   Female | 3014 | 55.86% | Respiration Rate (insp/min) | 19.37 | (15.00, 23.00) |
|   Male | 2382 | 44.14% | Systolic BP (mmHg) | 124.46 | (107.00, 141.00) |
| Age (yr) | 65.5 | (53.90, 79.93) | Diastolic BP (mmHg) | 66.75 | (55.00, 77.00) |
| Race/Ethnicity | | | Glasgow Comma Scale | 4.91 | (5.00, 6.00) |
|   Asian | 167 | 3.09% | Mechanical Ventilation | 2781 | 48.11% |
|   Black | 473 | 8.77% | Urine Output (ml) | 214.91 | (70.00, 300.00) |
|   Hispanic | 182 | 3.37% | Tidal Volume | 496.98 | (450.00, 550.00) |
|   White | 3923 | 72.70% | Temperature (celsius) | 36.64 | (36.10, 37.22) |
|   Other | 654 | 12.12% | History of Metastatic Cancer | 342 | 5.92% |
| Marital Status | | | History of Diabetes | 1629 | 28.18% |
|   Divorced | 328 | 6.08% | BMI | 28.54 | (23.53, 31.75) |
|   Married | 2379 | 44.09% | **Comprehensive Metabolic Panel ($48)** | | |
|   Single | 1536 | 28.47% | Glucose Level (mg/dL) | 4.91 | (4.65, 5.11) |
|   Widowed | 795 | 14.73% | Bicarbonate Level (mEq/L) | 22.95 | (20.00, 26.00) |
|   Unknown | 358 | 6.63% | Serum Creatinine (mg/dL) | 1.54 | (0.80, 1.60) |
| Admission Type | | | Aspartate aminotransferase (IU/L) | 4.01 | (3.18, 4.49) |
|   Elective | 310 | 5.74% | Bilirubin (mg/dL) | 1.57 | (0.40, 1.30) |
|   Emergency | 5026 | 93.14% | Chloride (mEq/L) | 105.29 | (101.00, 109.00) |
|   Urgent | 60 | 1.11% | Carbon Dioxide (mEq/L) | 24.61 | (21.00, 28.00) |
| Insurance Type | | | Lactate (mmol/L) | 2.16 | (1.10, 2.60) |
|   Government | 157 | 2.91% | Magnesium (mg/dL) | 1.92 | (1.70, 2.10) |
|   Medicaid | 532 | 9.86% | Blood Urea Nitrogen Level (mg/dL) | 28.85 | (14.00, 35.00) |
|   Medicare | 3123 | 57.88% | Blood albumin (g/dL) | 3.01 | (2.60, 3.40) |
|   Private | 1538 | 28.50% | Blood sodium (mEq/L) | 138.3 | (135.00, 141.00) |
|   Self-pay | 46 | 0.85% | Potassium Level (mEq/L) | 4.18 | (3.70, 4.60) |
| **Arterial Blood Gas Test ($473)** | | | **Complete Blood Count Panel ($44)** | | |
| Base excess (mEq/L) | -1.51 | (-4.00, 1.00) | Hemoglobin Level (g/dL) | 10.83 | (9.40, 12.20) |
| PH (unit) | 7.36 | (7.31, 7.43) | Hematocrit (%) | 32.37 | (28.30, 36.30) |
| Fraction of Inspired Oxygen (%) | 72.01 | (50.00, 100.00) | Platelet Count (K/$\mu$L) | 5.21 | (4.93, 5.59) |
| Mean Arterial Pressure (mmHg) | 82.08 | (69.00, 93.00) | Red Blood Cell Count (K/$\mu$L) | 3.58 | (3.10, 4.05) |
| Blood Oxygen Saturation (%) | 96.73 | (95.00, 100.00 | White Blood Cell Count (K/$\mu$L) | 12.69 | (7.50, 15.30) |
| **Composite Score** | | | **Activated Partial Thromboplastin Time Test Panel ($26)** | | |
| Sequential Organ Failure Assessment | 4.63 | (2.00, 6.00) | Partial Thromboplastin Time (s) | 36.75 | (26.50, 37.30) |
| **Outcome** | | | International Normalized Ratio | 1.5 | (1.10, 1.50) |
| In-hospital Mortality | 3945 | 16.47% | | | |

configurations can be tested simultaneously. Each training job requires a wall-time less than 2 hours of a single CPU core.

**Hyper-Parameters Settings**     The hyper-parameter of our model is divided onto four parts: parameter for training imputer, classifier, PPO policy, as well as the end-to-end training parameters. We list all the hyper-parameters we tuned in Table 7, including both the tuning range and final selection. The unmentioned parameters in training EMFlow imputer are set to the same as in its original work Ma & Ghosh (2021). The unmentioned parameters in PPO are set to default values in Python package Raffin et al. (2021).

**Data Splitting Scheme**     We split each dataset into 3 parts: training set (75%), validation set (15%), and test set (10%). Training data is further splitted into two disjoint sets, one for pretraining the state encoder (25%) and the other one for end-to-end RL training (50%). The validation set is splitted in the same way, one for tuning hyper-parameters for training the state encoder (5%) and the other one for tuning hyper-parameters for end-to-end RL training (10%). Here, all percentages are calculated w.r.t. the whole dataset size.

**Policy Improvement During RL Training**     In Figure 5, we showed the performance of the RL policies evaluated on train, validation and test sets. We note that the three curves closely match one another, confirming the generalizability of the learned dynamic classification policy.

**Zoom-in Version of the cost-$F_1$ Pareto Front**     Here we present the cost-$F_1$ Pareto front, comparing only to baselines with similar testing cost (Figure 6 is a subset of Figure 4). The yellow curve, which is the envelope of the 190 solutions, is the final Pareto front we obtained on the test set. Comparing to baselines with similar cost, our approach gives a better performance on $F_1$ score under the same cost budget. As already shown in Figure 4, even comparing to fully observable approaches which have

Table 6: Complete version of Table 2. Model performance, measured by $F_1$ score, area under ROC (AUC), and testing cost, for three real-world clinical datasets. The tested models include logistic regression (LR), random forests (RF), gradient boosted regression trees (XGBoost Chen & Guestrin (2016) and LightGBM Ke et al. (2017) ), 3-layer multi-layer perceptron, Q-learning for classification with costly features (CWCF)Janisch et al. (2019), Random Selection (RS), Fixed Selection (FS). Among them, classical methods including LR, RF, XGBoost, Lightbgm are tested under both fully observable case and fixed observable case where 2 most relavant panels (CMB and CBCP panel) are chosen. All models were fine-tuned to maximize the $F_1$ score. The model yielded the highest $F_1$ score is bold. The model required the least testing cost is underlined.

| Dataset | Models | Test Selection Strategy | $F_1$ | AUC | Cost |
|---|---|---|---|---|---|
| **Ferritin** | LR | Full | $0.541 \pm 0.010$ | $0.934 \pm 0.008$ | $290 |
| | RF | Full | $0.597 \pm 0.024$ | $0.931 \pm 0.006$ | $290 |
| | XGBoost | Full | $0.610 \pm 0.027$ | $0.940 \pm 0.004$ | $290 |
| | Lightbgm | Full | $\mathbf{0.625 \pm 0.011}$ | $\mathbf{0.941 \pm 0.006}$ | **$290** |
| | 3-layer DNN | Full | $0.615 \pm 0.018$ | $0.937 \pm 0.005$ | $290 |
| | LR (2 panels) | Fixed | $0.401 \pm 0.016$ | $0.859 \pm 0.013$ | $92 |
| | RF (2 panels) | Fixed | $0.504 \pm 0.023$ | $0.887 \pm 0.012$ | $92 |
| | XGBoost (2 panels) | Fixed | $0.519 \pm 0.015$ | $0.895 \pm 0.007$ | $92 |
| | Lightbgm (2 panels) | Fixed | $0.571 \pm 0.010$ | $0.901 \pm 0.006$ | $92 |
| | FS | Fixed | $0.588 \pm 0.017$ | $0.922 \pm 0.007$ | $74 |
| | RS | Random | $0.427 \pm 0.023$ | $0.837 \pm 0.012$ | $144.78 \pm 0.52$ |
| | CWCF | Dynamic | $0.531 \pm 0.021$ | $0.702 \pm 0.018$ | $261.02 \pm 1.20$ |
| | SM-DDPO$_{pretrained}$ | Dynamic | $0.587 \pm 0.021$ | $0.922 \pm 0.008$ | $80.72 \pm 0.63$ |
| | SM-DDPO$_{end2end}$ | Dynamic | $\underline{0.613 \pm 0.018}$ | $0.926 \pm 0.006$ | $\underline{62.51 \pm 0.46}$ |
| **AKI** | LR | Full | $0.453 \pm 0.028$ | $0.791 \pm 0.018$ | $591 |
| | RF | Full | $0.438 \pm 0.032$ | $0.771 \pm 0.016$ | $591 |
| | XGBoost | Full | $0.414 \pm 0.023$ | $0.789 \pm 0.017$ | $591 |
| | Lightbgm | Full | $0.474 \pm 0.017$ | $0.791 \pm 0.012$ | $591 |
| | 3-layer DNN | Full | $0.492 \pm 0.024$ | $0.797 \pm 0.010$ | $591 |
| | LR (2 panels) | Fixed | $0.473 \pm 0.006$ | $0.797 \pm 0.005$ | $92 |
| | RF (2 panels) | Fixed | $0.425 \pm 0.016$ | $0.768 \pm 0.004$ | $92 |
| | XGBoost (2 panels) | Fixed | $0.410 \pm 0.022$ | $0.781 \pm 0.009$ | $92 |
| | Lightbgm (2 panels) | Fixed | $0.491 \pm 0.011$ | $0.792 \pm 0.006$ | $92 |
| | FS | Fixed | $0.432 \pm 0.028$ | $0.789 \pm 0.015$ | $98 |
| | RS | Random | $0.411 \pm 0.029$ | $0.744 \pm 0.019$ | $294.88 \pm 1.32$ |
| | CWCF | Dynamic | $0.278 \pm 0.053$ | $0.510 \pm 0.009$ | $340.39 \pm 19.01$ |
| | SM-DDPO$_{pretrained}$ | Dynamic | $\mathbf{0.514 \pm 0.030}$ | $\mathbf{0.788 \pm 0.022}$ | $\mathbf{\underline{90.12 \pm 0.68}}$ |
| | SM-DDPO$_{end2end}$ | Dynamic | $0.489 \pm 0.026$ | $0.796 \pm 0.019$ | $97.31 \pm 0.58$ |
| **Sepsis** | LR | Full | $0.507 \pm 0.035$ | $0.814 \pm 0.030$ | $591 |
| | RF | Full | $0.442 \pm 0.051$ | $0.820 \pm 0.016$ | $591 |
| | XGBoost | Full | $0.451 \pm 0.051$ | $0.842 \pm 0.025$ | $591 |
| | Lightbgm | Full | $0.501 \pm 0.046$ | $0.850 \pm 0.018$ | $591 |
| | 3-layer DNN | Full | $0.515 \pm 0.031$ | $0.842 \pm 0.014$ | $591 |
| | LR (2 panels) | Fixed | $0.488 \pm 0.021$ | $0.811 \pm 0.015$ | $92 |
| | RF (2 panels) | Fixed | $0.478 \pm 0.053$ | $0.828 \pm 0.018$ | $92 |
| | XGBoost (2 panels) | Fixed | $0.459 \pm 0.032$ | $0.877 \pm 0.036$ | $92 |
| | Lightbgm (2 panels) | Fixed | $0.502 \pm 0.031$ | $0.864 \pm 0.010$ | $92 |
| | FS | Fixed | $0.489 \pm 0.039$ | $0.830 \pm 0.020$ | $90 |
| | RS | Random | $0.475 \pm 0.062$ | $0.779 \pm 0.030$ | $295.01 \pm 2.41$ |
| | CWCF | Dynamic | $0.118 \pm 0.011$ | $0.503 \pm 0.001$ | $401.83 \pm 120.76$ |
| | SM-DDPO$_{pretrained}$ | Dynamic | $\mathbf{0.545 \pm 0.034}$ | $\mathbf{0.834 \pm 0.022}$ | $\mathbf{\underline{86.21 \pm 1.44}}$ |
| | SM-DDPO$_{end2end}$ | Dynamic | $0.532 \pm 0.036$ | $0.841 \pm 0.023$ | $90.55 \pm 1.20$ |

much higher testing cost, our approach exhibits comparing performance on $F_1$ score and AUROC with a much lower testing cost.

Lastly, we emphasize that our approach gives the whole Pareto front, which represents a complete and rigorous characterization of the accuracy-cost trade-off in our medical diagnostics tasks. While all the baselines we compared to, only gives single points on this trade-off characterization.

Table 7: Hyper-parameter Table

| Hyper-parameters | Ferritin | AKI | Sepsis | Selection Range |
|---|---|---|---|---|
| **_Imputer_** | | | | |
| Batch size | 256 | 256 | 256 | {64, 128, 256} |
| Learning rate | 1e-3 | 1e-4 | 1e-4 | {1e-3, 1e-4} |
| Regularization $\alpha$ | 1e3 | 1e6 | 1e6 | {1, 1e3, 1e6} |
| # Iterations | 500 | 500 | 500 | {100, 300, 500} |
| **_Classifier_** | | | | |
| Hidden size (3-layer) | 64 | 256 | 256 | {64, 128, 256} |
| Batch size | 256 | 256 | 256 | {64, 128, 256} |
| Learning rate | 5e-4 | 1e-4 | 1e-4 | {1e-3, 5e-4, 1e-4, 5e-5, 1e-5} |
| Class weight | 3 | 5 | 5 | {1,3,5,8,10,12} |
| **_PPO_** | | | | |
| Learning rate | 1e-4 | 1e-4 | 1e-4 | {1e-3, 1e-4} |
| Hidden size (2-layer) | 128 | 256 | 128 | {128, 256} |
| Batch size | 128 | 128 | 256 | {64, 128, 256} |
| # Timesteps per update | 1024 | 2048 | 2048 | {1024, 2048} |
| **_End-to-end_** | | | | |
| # Outer loop | 100 | 100 | 4 | {10,100} |
| # Epochs Classifier trained per loop | 6 | 8 | 6 | {2,4,6,8} |
| # Timesteps PPO trained per loop | 5k | 50k | 50k | {5k, 50k} |

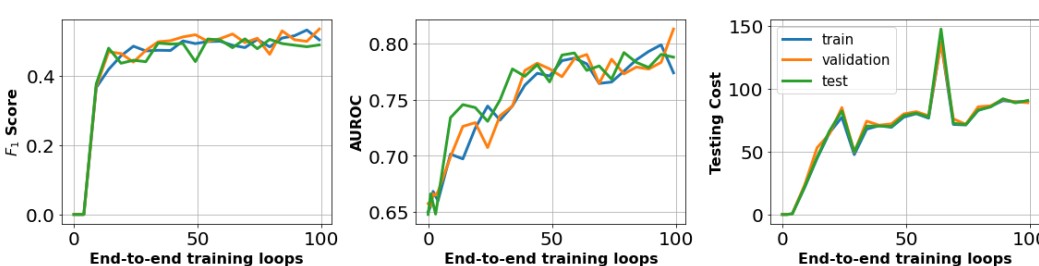

Figure 5: RL training curves on AKI Dataset. We evaluate the policy and classifier learned during the end-to-end training phase on both train, validation and test set. The matching curve confirms generalizability of the learned policies.

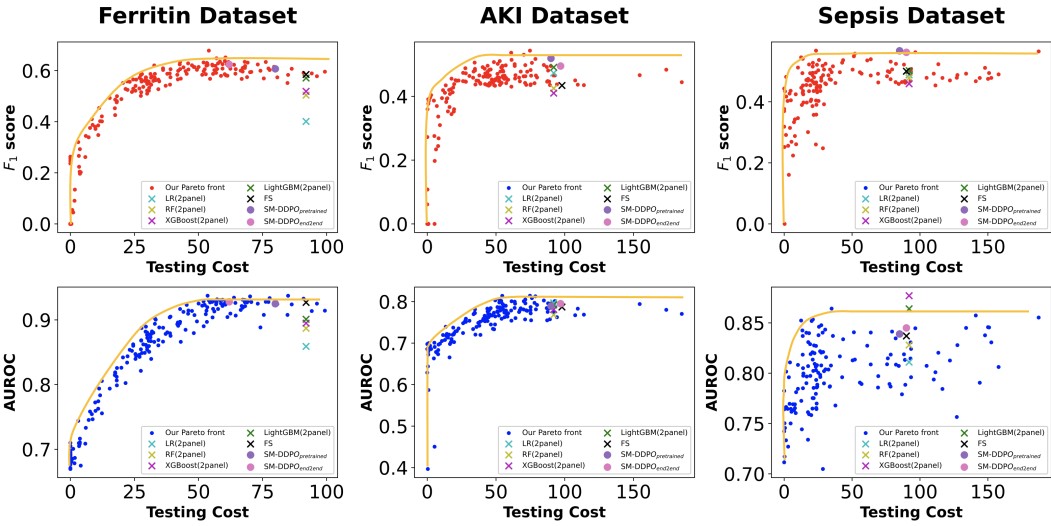

Figure 6: Cost-$F_1$ Pareto Front for maximizing F1-score on Ferritin, AKI and Sepsis Datasets

## C  EMFLOW

In this section, we describe the EMFlow imputation algorithm, originally proposed in Ma & Ghosh (2021), used in constructing the posterior state encoder. The key idea of EMFlow is to first map the data onto the latent space via a normalizing flow model, such that it enjoys a Gaussian distribution in the latent space. Then it apply an online version of the expectation maximization algorithm in the latent space. The high-level picture of EMFlow is illustrated in Fig. 7. In each training iteration, we

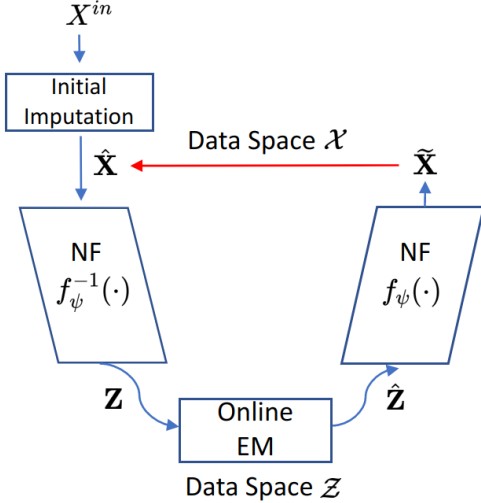

Figure 7: Illustration of EMFlow Ma & Ghosh (2021).

first fix the estimate base distribution $\mathcal{N}(\hat{\mu}, \hat{\Sigma})$ in the latent space and update the normalizing flow (NF) via optimizing the negative log-likelihood of a batch $B$ of the current imputation result:

$$L_1(\theta) = -\frac{1}{|B|} \sum_{i \in B} \log p_X(\hat{\mathbf{x}}_i; \theta, \hat{\mu}, \hat{\Sigma}) \tag{4}$$

where NF is parametrized by $\theta$ and $p_X$ denotes the likelihood of the current imputation estimate $\mathbf{x}_i$ given the base distribution. Then with the current estimate of the normalizing flow, we do an online version of expectation maximization (EM) in the latent space to update the estimate for the base distribution. The expectation step is given as:

$$\hat{\mathbf{z}}_i^m = \mathbb{E}[\hat{\mathbf{z}}_i^m | \hat{\mathbf{z}}_i^o; \hat{\mu}, \hat{\Sigma}], \ i \in B \tag{5}$$

And the maximization step follows an iterative update scheme to deal with the memory overhead of the classical EM:

$$\hat{\mu}^{(t)} = \rho_t \hat{\mu}_{batch} + (1 - \rho_t)\hat{\mu}^{(t-1)} \tag{6}$$

$$\hat{\Sigma}^{(t)} = \rho_t \hat{\Sigma}_{batch} + (1 - \rho_t)\hat{\Sigma}^{(t-1)} \tag{7}$$

where $\hat{\mu}_{batch}$ and $\hat{\Sigma}_{batch}$ are estimated using the conditional mean and variance given a batch of data:

$$\hat{\mu}_{batch} = g_\mu(\hat{\mu}^{(t)}, \hat{\Sigma}^{(t)}; \{\mathbf{z}_i, M_i\}_{i \in B}) \tag{8}$$

$$\hat{\Sigma}_{batch} = g_\Sigma(\hat{\mu}^{(t)}, \hat{\Sigma}^{(t)}; \{\mathbf{z}_i, M_i\}_{i \in B}) \tag{9}$$

Lastly, we re-optimize NF via a combination of log-likelihood of current imputation and a regularization term indicating the distance of the imputation result to the ground truth.

$$L_2(\theta) = -\frac{1}{|B|} \sum_{i \in B} [\log p_X(\tilde{\mathbf{x}}_i; \theta, \hat{\mu}, \hat{\Sigma}) - \alpha \|\tilde{\mathbf{x}}_i - \mathbf{x}_i\|_2^2] \tag{10}$$

In this work, we used the above "supervised" version of EMFlow, as we already know the ground truth data when training this imputer. Finally, the imputation procedure is in similar manner as

---

**Algorithm 2** Training Phase of EMFlow ("supervised" version)

---

**Input:** Current imputation $\hat{\mathbf{X}} = (\hat{\mathbf{x}}_1, \hat{\mathbf{x}}_2, \cdots, \hat{\mathbf{x}}_n)^\top$, missing masks $\mathbf{M} = (M_1, M_2, \cdots, M_n)^\top$, initial estimates of the base distribution $(\hat{\mu}^{(0)}, \hat{\Sigma}^{(0)})$, online EM step size sequence: $\rho_1, \rho_2, \cdots$,
**for** $t = 1, 2, \cdots, T_{\text{epoch}}$ **do**

    Get a mini-batch $\hat{\mathbf{X}}_B = \{\hat{\mathbf{x}}_i\}_{i \in B}$
    # update the flow model
    Compute $L_1$ in Eq. (4)
    Update $\theta$ via gradient descent
    # update the base distribution
    $\mathbf{z}_i = f_\theta^{-1}(\hat{\mathbf{x}}_i), i \in B$
    Impute in the latent space with $(\hat{\mu}^{(t-1)}, \hat{\Sigma}^{(t-1)})$ to get $\{\mathbf{z}_i\}_{i \in B}$ via Eq. (5)
    Obtain updated $(\hat{\mu}^{(t)}, \hat{\Sigma}^{(t)})$ via Eq. (6)
    # update the flow model again
    $\tilde{\mathbf{x}}_i = f_\theta(\hat{\mathbf{z}}_i), i \in B$
    Compute $L_2$ via Eq. (10)
    Update $\theta$ via gradient descent
**end for**

---

**Algorithm 3** Re-imputation Phase of EMFlow

---

**Input:** Current imputation $\hat{\mathbf{X}} = (\hat{\mathbf{x}}_1, \hat{\mathbf{x}}_2, \cdots, \hat{\mathbf{x}}_n)^\top$, missing masks $\mathbf{M} = (M_1, M_2, \cdots, M_n)^\top$, initial estimates of the base distribution $(\hat{\mu}, \hat{\Sigma})$ from the previous training phase
**for** $t = 1, 2, \cdots, n$ **do**

    $\mathbf{z}_i = f_\theta^{-1}(\hat{\mathbf{x}}_i), i \in B$
    Impute in the latent space with $(\hat{\mu}^{(t-1)}, \hat{\Sigma}^{(t-1)})$ to get $\{\mathbf{z}_i\}_{i \in B}$ via Eq. (5)
    $\tilde{\mathbf{x}}_i = f_\theta(\hat{\mathbf{z}}_i), i \in B$
    $\hat{\mathbf{x}}_i = \hat{\mathbf{x}}_i \odot (1 - M_i) + \tilde{\mathbf{x}}_i \odot M_i$
**end for**
**Output:** $\{\hat{\mathbf{x}}_i\}_{i=1}^n$

---

the training phase without updating the parameters. Both the training and imputation scheme are illustrated in Algorithm 2 and 3 respectively.

To illustrate the imputation quality of the above EMFlow algorithm, we present a simple results in Table 8 of the root mean square error (RMSE) compared to classical imputation algorithms on UCI datasets as a reference. We do point out unlike conventional usage of imputation, we use EMFlow mainly as a posterior state encoder, instead of a imputer with excellent imputation quality.

Table 8: Comparison of Imputers on Public Datasets in RMSE. Each dataset has 20% MCAR. Apart from the time-consuming MissForest Imputer, the EMFlow outperforms other imputers in most datasets.

| Imputer | Letter | Spam | News | Air | Credit |
|---|---|---|---|---|---|
| Mean | 0.1535 | 0.0573 | 0.2265 | 0.1877 | 0.1592 |
| MICE | 0.1148 | 0.0664 | 0.1500 | 0.1014 | 0.1862 |
| MissForest | 0.0608 | 0.0476 | 0.1365 | 0.0869 | 0.1430 |
| Matrix Completion | 0.1290 | 0.0528 | 0.1653 | 0.1261 | 0.1632 |
| GAIN | 0.1198 | 0.0513 | 0.1441 | 0.1318 | 0.1858 |
| EMFlow | 0.1111 | 0.0776 | 0.1395 | 0.1022 | 0.1410 |

## D  END-TO-END TRAINING

In this section, we give more details of the proposed end-to-end training approach discussed in 5. We give out specific form for the objective of training both the classifier module and the panel/prediction selector module.

**Training objective $\ell_c$ for the classifier module**  The classifier $f_\phi(\cdot) : \mathbb{R}^d \to \mathbb{R}^C$, parameterized by $\phi$, maps the posterior encoded state $\mathrm{Imp}_\theta(\mathbf{x} \odot M)$ to a probability distribution over labels. It is trained by directly minimizing the cross entropy loss $\ell_c$ defined as:

$$\ell_c(\phi; \mathbf{x}, M) = -\sum_{i=1}^{2} w_i \log \frac{\exp f_\phi(\mathrm{Imp}_\theta(\mathbf{x} \odot M))_i}{\sum_{i=1}^{2} \exp f_\phi(\mathrm{Imp}_\theta(\mathbf{x} \odot M))_i}.$$

from the collected dataset $Q$ in training the RL policy. This differs from typical classification in that the data are collected adaptively by RL rather than sampled from a prefixed source. Here $w_i$'s are the class weights which treated as a tuning parameter. In this work, we used a 3-layer DNN as the model for the classifier.

**Training objective $\ell_{rl}$ for the panel/prediction selector module**  The panel selector, a network module parameterized by $\psi$, takes the following as input

$$s_{\theta,\phi}^{\mathrm{emb}}(s) = s_{\theta,\phi}^{\mathrm{emb}}(\mathbf{x} \odot M) = (\mathrm{Imp}_\theta(\mathbf{x} \odot M), f_\phi(\mathrm{Imp}_\theta(\mathbf{x} \odot M)), M), \quad\quad (11)$$

and maps it to a probability distribution over actions. We train the panel selector by using the classical proximal policy updates (PPO) Schulman et al. (2017), which aims at maximizing a clipped surrogate objective regularized by a square-error loss of the value functions and an entropy bonus. We denote this loss function as $\ell_{rl}$, which is defined as the following regularized clipped surrogate loss Schulman et al. (2017):

$$\ell_{rl}(\psi; \psi^{old}) := \ell_{CLIP}(\psi; \psi^{old}) - c_1 \ell_{VF}(\psi) + c_2 \mathrm{Ent}[\pi_\psi],$$

where $\ell_{CLIP}(\psi; \psi^{old}) = \hat{\mathbb{E}}_t \left[ \min(\frac{\pi_\psi(a_t|\hat{s}_t)}{\pi_{\psi^{old}}(a_t|\hat{s}_t)} \hat{A}_t, \mathrm{clip}(\frac{\pi_\psi(a_t|\hat{s}_t)}{\pi_{\psi^{old}}(a_t|\hat{s}_t)}, 1 - \epsilon, 1 + \epsilon)\hat{A}_t) \right]$ denotes the clipped surrogate loss with $\hat{A}_t$ being the estimated advantages, regularized by the square error loss of the value function $\ell_{VF}(\psi) = \hat{\mathbb{E}}_t[(V_\psi(\hat{s}_t) - V_t^{targ})^2]$ and an entropy term $\mathrm{Ent}[\pi_\psi]$ over states. Here $\hat{\mathbb{E}}_t$ is empirical average over collected dataset $Q$, and $\hat{s}_t = s_{\theta,\phi}^{\mathrm{emb}}(s_t)$ denotes the state embedding we derived. Note the policy and value network share the parameter $\psi$. We also show a high-level algorithm 4 here. For more details, please refer to the original paper Schulman et al. (2017). We use the Python package stable-baseline3 Raffin et al. (2021) for implementing PPO.

---

**Algorithm 4** Proximal Policy Optization (PPO)

---

**for** iteration = 0, 1, ⋯ **do**
    **for** actor = 1, 2, ⋯ , $N_{actor}$ **do**
        Run policy $\pi_{\psi^{old}}$ in environment for $T$ timesteps and save all observations in $Q$
                                        ▷ $Q$ is also used to train the classifier module
        Compute advantage estimate $\hat{A}_1, \cdots, \hat{A}_T$
    **end for**
    Optimizae surrogate $\ell_{rl}$ wrt $\psi$, with $K$ epochs and minibatch size $M \leq N_{actor}T$
    $\psi_{old} \leftarrow \psi$
**end for**

---

# E   PROOFS OF THEOREM 4.1

For the ease of reading, we present the Theorem 4.1 in Appendix again.

**Theorem E.1** (Copy of Theorem 4.1). *The Cost-$F_1$ Pareto front defined in* (1) *is a subset of the collection of all reward-shaped solutions, given by*

$$\Pi^* \subseteq \overline{\Pi} := \cup_{\lambda \geq 0, \rho \leq 0} \; \underset{\pi}{\mathrm{argmax}} \left\{ TN(\pi) + \lambda \cdot TP(\pi) + \rho \cdot Cost(\pi) \right\}.$$

As a natural corollary, we have the following result:

**Corollary E.1.** *The Cost-$F_1$ Pareto front defined in* (1) *is a subset of the solutions of the MDP model for dynamic diagnosis process defined in Section 3 with reward functions:*

$$R((s,a) = \begin{cases} \rho \cdot c(a), & \text{if } a \in [D] \text{ (choosing task panels)} \\ \lambda \cdot \mathbf{1}\{y = P\}, & \text{if } a = P \text{ (true positive diagnosis)} \\ \mathbf{1}\{y = N\}, & \text{if } a = N \text{ (true negative diagnosis)} \end{cases},$$

*for all pairs of* $(\lambda, \rho)$*. Here $y$ denotes the ground-truth label of the patient.*

Before presenting the proofs for Theorem 4.1, we reintroduce some key notations. First, we define the length of the MDP episode as $\tau = \min\{t \geq 0 | a_t \in \{P, N\}\}$ as a random variable depending on the policy $\pi$. Then we can rigorously define the normalized true positive, true negative, false positive and false negative as

$$\mathrm{TP}(\pi) := g(P, P), \; \mathrm{TN}(\pi) := g(N, N), \; \mathrm{FP}(\pi) := g(P, N), \; \mathrm{FN}(\pi) := g(N, P).$$

Here $g(i, j) := \mathbb{E}^\pi[\mathbf{1}\{a_\tau = i\} \cdot \mathbf{1}\{y = j\}]$ denotes the probability of the policy diagnosing class $i \in \{P, N\}$ while the ground truth label is of class $j \in \{P, N\}$. And the testing cost is defined as:

$$\mathrm{Cost}(\pi) = \mathbb{E}^\pi \left[ \sum_{t=0}^{\tau-1} \sum_{k \in [D]} c(k) \cdot \mathbf{1}\{a_t = k\} \right],$$

where $c(k)$ is the cost of panel $k$.

The cumulative state-action occupancy measure $\mu : \Delta_{\mathcal{A}}^{\mathcal{S}} \to \mathbb{R}_{\geq 0}^{\mathcal{S} \times \mathcal{A}}$, is defined as

$$\mu^\pi(s, a) := \mathbb{E}^\pi \left[ \sum_{t=0}^{\tau} \mathbf{1}(s_t = s, a_t = a) \right], \; \forall s \in \mathcal{S}, a \in \mathcal{A},$$

which denotes the expected time spent in state-action pair $(s, a)$ during an episode. There is a one-one correspondence between the policy and the occupancy measure given by

$$\pi(a|s) = \frac{\mu(s, a)}{\sum_{a \in \mathcal{A}} \mu(s, a)}.$$

*Proof of Theorem 4.1.* We follow the proof framework stated in the main text.

**Step 1: utilizing monotonicity of $F_1$ score**    To start with, we note that $F_1$ score is monotonically increasing in both true positive and true negative. Assume for any given cost budget $B$, the optimal policy $\pi^*(B)$ achieves the highest $F_1$ score. Then $\pi^*(B)$ is also the optimal to the following program:

$$\max_\pi \left\{ \mathrm{TN}(\pi) \text{ subject to } \mathrm{Cost}(\pi) \leq B, \mathrm{TP}(\pi) \geq \mathrm{TP}(\pi^*(B)) \right\},$$

indicating the Pareto front of $F_1$ score is a subset of

$$\Pi^* \subseteq \cup_{B>0, B' \in [0,1]} \operatorname*{argmax}_\pi \left\{ \mathrm{TN}(\pi) \text{ subject to } \mathrm{Cost}(\pi) \leq B, \mathrm{TP}(\pi) \geq B' \right\}. \tag{12}$$

**Step 2: reformulation using occupancy measures**    Fix any specific pair $(B, B')$. Consider the equivalent dual linear program form Zhang et al. (2020) of the above policy optimization problem in terms of occupancy measure. Then the program (12) is equivalent to:

$$\begin{aligned}
\max_\mu \quad & \mathrm{TN}(\mu) \\
\text{subject to} \quad & \mathrm{Cost}(\mu) \leq B, \\
& \mathrm{TP}(\mu) \geq B', \\
& \sum_a \mu(s,a) = \sum_{s',a' \in [D]} \mu(s',a') P(s|s',a') + \xi(s), \forall s,
\end{aligned}$$

where $\xi(\cdot)$ denotes initial distribution, and TP, TN and cost are reloaded in terms of occupancy $\mu$ as:

$$\mathrm{TP}(\mu) = \sum_{s.y=\mathrm{P}, a=\mathrm{P}} \mu(s,a), \ \mathrm{TN}(\mu) = \sum_{s.y=\mathrm{N}, a=\mathrm{N}} \mu(s,a), \ \mathrm{Cost}(\mu) = \sum_{k \in [D]} c(k) \cdot \sum_{s,a=k} \mu(s,a).$$

**Step 3: utilizing hidden minimax duality**    The above program can be equivalently reformulated as a max-min program:

$$\begin{aligned}
\max_\mu \min_{\lambda \geq 0, \rho \leq 0} \quad & \mathrm{TN}(\mu) + \lambda \cdot (\mathrm{TP}(\mu) - B') + \rho \cdot (\mathrm{Cost}(\mu) - B) \\
\text{subject to} \quad & \sum_a \mu(s,a) = \sum_{s',a' \in [D]} \mu(s',a') P(s|s',a') + \xi(s), \forall s.
\end{aligned}$$

Note the max-min objective is linear in terms of $\lambda, \rho$ and $\mu$. Thus minimax duality holds, then we can swap the min and max to obtain the equivalent form:

$$\begin{aligned}
\min_{\lambda \geq 0, \rho \leq 0} \max_\mu \quad & \mathrm{TN}(\mu) + \lambda \cdot (\mathrm{TP}(\mu) - B') + \rho \cdot (\mathrm{Cost}(\mu) - B) \\
\text{subject to} \quad & \sum_a \mu(s,a) = \sum_{s',a' \in [D]} \mu(s',a') P(s|s',a') + \xi(s), \forall s.
\end{aligned}$$

For any fixed pair of $(\lambda, \rho)$, the inner maximization problem of the above can be rewritten back in terms of policy $\pi$, as the following equivalent unconstrained policy optimization problem:

$$\max_\pi \ \mathrm{TN}(\pi) + \lambda \cdot \mathrm{TP}(\pi) + \rho \cdot \mathrm{Cost}(\pi).$$

This is finally a standard cumulative-sum MDP problem, with reshaped reward: reward $\rho \cdot c(t)$ for the action of choosing test panel $t$, reward $\lambda$ for the action to diagnose and get a true positive, reward 1 for getting a true negative, as stated in Corollary E.1:

$$R((s,a)) = \begin{cases} \rho \cdot c(a), & \text{if } a \in [D] \text{ (choosing task panels)} \\ \lambda \cdot \mathbf{1}\{y = \mathrm{P}\}, & \text{if } a = \mathrm{P} \text{ (true positive diagnosis)} \\ \mathbf{1}\{y = \mathrm{N}\}, & \text{if } a = \mathrm{N} \text{ (true negative diagnosis)} \end{cases}.$$

$\square$

# F    EXTENSION TO AM METRIC

In this section, we show that our framework directly handles classic linear metrics defined as a linear combination of (normalized) true positives and true negatives. We use the AM metric Natarajan et al. (2018); Menon et al. (2013), a linear metric for imbalanced tasks via considering the average of true positive rate and true negative rate, as an example.

We can formally define the cost-AM Pareto front as follows:

$$\Pi^*_{AM} = \cup_{B>0} \; \underset{\pi}{\mathrm{argmax}} \{AM(\pi) \text{ subject to } \mathrm{Cost}(\pi) \le B\}$$

where $AM(\pi) = \frac{1}{2}(\mathrm{TPR}(\pi) + \mathrm{TNR}(\pi))$. Here we define the true positive rate and true negative rate respectively as:

$$\mathrm{TPR}(\pi) := \frac{\mathrm{TP}(\pi)}{\mathrm{TP}(\pi) + \mathrm{FN}(\pi)}, \quad \mathrm{TNR}(\pi) := \frac{\mathrm{TN}(\pi)}{\mathrm{TN}(\pi) + \mathrm{FP}(\pi)},$$

where $\mathrm{TP}(\pi), \mathrm{TN}(\pi), \mathrm{FP}(\pi), \mathrm{FN}(\pi)$ are normalized true positive, true negative, false positive and false negative rates that sum up to 1. Let $\lambda > 0$ be the **known** ratio between the number of healthy patients and ill patients. Then we have $\mathrm{TP}(\pi) + \mathrm{FN}(\pi) = \frac{1}{1+\lambda}$ and $\mathrm{TN}(\pi) + \mathrm{FP}(\pi) = \frac{\lambda}{1+\lambda}$, indicating

$$AM(\pi) = \frac{1+\lambda}{2\lambda}(\lambda \mathrm{TP}(\pi) + \mathrm{TN}(\pi))$$

Thus, we show the linearity of AM metric, indicating given any fixed budget, the problem already reduced back to a standard MDP. It is a simpler problem and we can directly solve them using our training framework. Rigorously, we have the following result parallel to our Theorem 1:

$$\Pi^*_{AM} = \cup_{\rho \le 0} \; \underset{\pi}{\mathrm{argmax}} \{\mathrm{TN}(\pi) + \lambda \cdot \mathrm{TP}(\pi) + \rho \cdot \mathrm{Cost}(\pi)\}.$$

Empirically, we show the following Table 9 of our results on AM metric, comparing to the same baselines discussed in our work. The testing costs are similar to the Table 2 in the paper. It is shown that our approach achieves the best AM score while having lower testing costs.

Table 9: Model performance measured by balanced accuracy for three real-world clinical dataset. The model yielded the highest balanced accuracy is in bold.

| Models | AKI | | Ferritin | | Sepsis | |
|---|---|---|---|---|---|---|
| *Metrics* | *AM* | *Cost* | *AM* | *Cost* | *AM* | *Cost* |
| LR | 0.714 | $591 | 0.842 | $290 | 0.752 | $591 |
| RF | 0.668 | $591 | 0.817 | $290 | 0.722 | $591 |
| XGBoost | 0.644 | $591 | 0.763 | $290 | 0.644 | $591 |
| LightGBM | 0.690 | $591 | 0.832 | $290 | 0.751 | $591 |
| LR (2 panels) | 0.714 | $92 | 0.783 | $92 | 0.746 | $92 |
| RF (2 panels) | 0.660 | $92 | 0.778 | $92 | 0.707 | $92 |
| XGBoost (2 panels) | 0.639 | $92 | 0.683 | $92 | 0.642 | $92 |
| LightGBM (2 panels) | 0.688 | $92 | 0.786 | $92 | 0.737 | $92 |
| **SM-DDPO** | **0.741** | $92 | **0.845** | $95 | **0.754** | $48 |

# G    PRECISION AND RECALL

We also show the detailed statistics of two sets of optimal solutions that optimizes F1 score and AM metric respectively. It can be shown by comparing Table 10 and 11 that F1 score balances the recall and precision while AM metric balances the true positive and negative rates.

Table 10: Detailed statistics of our results optimizing F1 score.

| Dataset | Recall | Precision | F1 | true positive rate | true negative rate | AM |
|---|---|---|---|---|---|---|
| Ferritin | 0.664 | 0.587 | 0.624 | 0.664 | 0.954 | 0.809 |
| AKI | 0.537 | 0.459 | 0.495 | 0.537 | 0.875 | 0.706 |
| Sepsis | 0.573 | 0.551 | 0.562 | 0.573 | 0.921 | 0.747 |

Table 11: Detailed statistics of our results optimizing AM metric.

| Dataset | Recall | Precision | F1 | true positive rate | true negative rate | AM |
|---|---|---|---|---|---|---|
| Ferritin | 0.798 | 0.420 | 0.550 | 0.798 | 0.892 | 0.845 |
| AKI | 0.704 | 0.385 | 0.498 | 0.704 | 0.778 | 0.741 |
| Sepsis | 0.613 | 0.497 | 0.549 | 0.613 | 0.895 | 0.754 |

