# OpenReview forum: "Deep Reinforcement Learning for Cost-Effective Medical Diagnosis"
_ICLR.cc/2023/Conference — ICLR 2023 poster_

### Official Review · Reviewer_i4nR · 2022-10-23

**Confidence:** 3
**Correctness:** 3
**Technical Novelty And Significance:** 3
**Empirical Novelty And Significance:** 3
**Recommendation:** 6

**Clarity, Quality, Novelty And Reproducibility:**

Quality: I think the paper is of good quality. The authors justify the claims in the paper through their problem and method formulation as well as empirical results.

Clarity: I think the paper is well written, and the major arguments made by the paper can be followed easily. One particular claim that is not quite clear to me is the multi-objective formulation of the problem in (1). It seems to me this is a set of constrained optimization problems under various cost budgets rather than multiobjective?

Originality: I think the problem that the paper seeks to address is new and interesting. The paper offers a sensible approach to formulating and solving the problem.

**Strength And Weaknesses:**

Strength:
1. the paper deals with an important problem in machine learning for health care.
2. The paper proposes a semi-model-based RL approach to learning the diagnostic policy, which seems suitable to the problem formulation.
3. Experiments are carried out on three real-world datasets, which is a fair amount of different problem settings. The proposed method is compared to 12 alternatives, which is fairly exhaustive.

Weakness:
1. It is not clear to me how necessary it is to directly optimize for F1 via reward shaping, given that there are alternative linear metrics available for imbalanced classification such as the AM metric. See, for examples,
https://jmlr.org/papers/v18/15-226.html
https://proceedings.mlr.press/v28/menon13a.html

2. It is not clear to me how interpretable the diagnostic policy might be. This is in contrast to the tree-based baseline methods compared to the proposed methods.



**Summary Of The Paper:**

The paper proposed a reinforcement learning-based approach for test panel selection in order to make diagnostic predictions under cost constraints. The paper deals with data imbalance in the data by a rewarding shaping approach. It proposed a semi-model-based approach to learning the diagnostic policy. Experiments are carried out on three real-world datasets to demonstrate the utility of the proposed method.

**Summary Of The Review:**

Overall, I am leaning toward accepting the paper because the paper deals with an interesting problem in ML for health, formulates and develops a reasonable solution to the problem, and evaluates the proposed method rigorously across multiple real-world datasets over multiple baseline methods.

My concerns are centered around the necessity of using the F1 metric, the interpretability of the proposed method given that the methods address problems in the healthcare domain, and the multi-objective formulation of the problem. It may also be desirable for the authors to discuss the applicability of alternative metrics for imbalanced classification such as the AM metrics and potentially carry out empirical comparison.

---

### Official Review · Reviewer_Ws9d · 2022-10-23

**Confidence:** 3
**Correctness:** 3
**Technical Novelty And Significance:** 3
**Empirical Novelty And Significance:** 2
**Recommendation:** 8

**Clarity, Quality, Novelty And Reproducibility:**

**Clarity:** This paper is largely well written.

**Quality:** The proposed method is accompanied with high-level summary of the main theoretical results and detailed derivations in appendix, and validated on three real-world clinical datasets.

**Novelty:** Based on the background of this reviewer, this work presents an interesting solution for active feature acquisition that incorporates classification metrics other than accuracy (here the F1-score). I am not too familiar with the literature of this area so cannot comment on the extensiveness of related works.

**Reproducibility:** Code is provided as part of the supplement and looks like a good-faith effort of showing commitment of open-sourcing from by the authors.

**Strength And Weaknesses:**

**Strengths:**
- The problem studied by this paper is interesting and important for many clinical classification problems subject to various budget constraints.
- Novel theoretical insights on the how the optimization problem can be transformed to formulate a reward function that is an average of episode rewards.
- Empirical results are strong, the proposed approach shows favorable performance on three different datasets.

**Weaknesses:**
- The title is missing keywords such as "cost-sensitive" and "active feature acquisition" ("diagnostic policy learning" might be a new word to many readers).
- In Fig 5, some of the red/blue points lie outside the Pareto front. How are the Pareto front curves fitted?
- Some statements could use a bit more clarification:
  - in Step 3, $(s',a')$ with the apostrophe is typically used to indicate next state-action pair, where here in the occupancy definition they represent the previous state-action pair. It is suggested that the authors change the notation here to something else like $(\tilde{s}, \tilde{a})$ throughout.
  - in the abstract, "the F1 score cannot be formulated as a simple sum of cumulative rewards" - on first read this sentence is confusing, it seems to be discussing the cumulative rewards of a single trajectory, but the F1-score is ill defined at the trajectory/instance level. Only after reading the paper did I understand it's actually talking about the sum (or average) of multiple trajectories. I suggest this sentence be rephrased.
  - on page 4, Step 2: "Fix any specific pair (B,B')" should be "Fix any specific pair (B,K)"
  - in Sec 4 the definition of TP(\pi) vs TP(\mu) seems inconsistent on their normalization. Are they the number of true positives, or the true positive rate?
  - in Sec 3.1 and Fig 1 "A penalty will [be] generated if the diagnosis does not match the ground truth $y$." This sentence is inconsistent with Sec 4 Step 3 reward definition $\mathrm{TN} + \lambda \mathrm{TP} + \rho \mathrm{Cost}$ where no penalty is assigned, only positive rewards for true pos/neg.
  - On page 5: regarding "semi-model-based", could you clarify why the reward needs to be learned? It seems that the reward r(s,a) should only depend on the diagnosis prediction and the true label.
  - Sec 3.1 should also clarify that the transition dynamics of the MDP can be perfectly specified.
   - in Sec 5.3: Eqn (3) is not only the panel selector, it's panel/prediction selector which is the policy network.

**Summary Of The Paper:**

This paper studies sequential selection of laboratory test panels, formulated as a cost-sensitive adaptive feature selection/acquisition problem. The two main innovations are:
1. For diagnostic performance, instead of maximizing accuracy, this work proposes to maximize F1 score to address high class imbalance (how high ). Since the F1 score cannot be written as an average of episode returns and thus cannot be directly formulated as a reward function, the paper presents new theoretical insights on transformations of the optimization problem which led to a reward shaping approach that allows RL to be applied.
2. Viewing the problem as multi-objective optimization of diagnostic performance and cost, the proposed reward function is able to identify a superset of all Pareto optimal solutions, allowing one to recover the Pareto front of optimal F1 for each cost.

Experiments were conducted on real-world datasets - ferritin, sepsis and AKI - and achieves good diagnostic performance while having lower costs than competitors.

**Summary Of The Review:**

The problem studied is interesting and relevant, the proposed solution makes sense. Certain claims need refinement and experiments need some clarifications.

---

### Official Review · Reviewer_Yh4E · 2022-10-25

**Confidence:** 3
**Correctness:** 3
**Technical Novelty And Significance:** 3
**Empirical Novelty And Significance:** 3
**Recommendation:** 6

**Clarity, Quality, Novelty And Reproducibility:**

The presentation is clear. The proposed method is novel, as far as I know. The proposed method is also sound to me.

**Strength And Weaknesses:**

Strength:
1. The studied problem is very interesting and practically useful, and I think using reinforcement learning for lab test panel optimization is an appropriate way.
2. The proposed method seems sound to me (but I did not check the proofs).
3. The writing is good.

Weakness:
1. The reported empirical performance of the proposed method has an improvement in most cases, compared to other methods. However, the F1 score still seems pretty low, which may still not be appropriate for clinical use.

2. I would suggest the authors pay attention to the format, avoiding using too many \vspace{}; some parts can be moved to the appendix.

**Summary Of The Paper:**

This paper investigates the use of reinforcement learning for lab test panel optimization, aiming to dynamically prescribe test panels based on available observations, to maximize diagnosis/prediction accuracy while keeping testing at a low cost. Given that clinical diagnostic data are often highly imbalanced, the authors maximize the F1 score instead of the error rate, and further develop a reward-shaping approach by leveraging the duality of policy optimization, so that the problem can be solved by standard RL methods.

**Summary Of The Review:**

The presentation is clear. The proposed method is novel and sound to me. Although the empirical performance is better than other methods in most cases, the F1 score still seems pretty low, which may still not be appropriate for clinical use.

---

### Decision · Program_Chairs · 2023-01-20

**Decision:**

Accept: poster

**Justification For Why Not Higher Score:**

The paper is interesting but there are also no surprises/fundamental results presented.

**Justification For Why Not Lower Score:**

The paper makes some interesting contributions and the approach developed to maximize the F1 score is interesting. Hence, I think, the community would benefit from seeing this paper published.

**Metareview: Summary, Strengths And Weaknesses:**

This paper considers the problem of maximizing the F1 score in an RL setting motivated by applications for cost-sensitive diagnostic planning. The authors develop a reward-shaping approach for finding Pareto-optimal policies for the case of budgeted F1-score maximization. Furthermore, they empirically demonstrate that their approach works well on several real-world datasets.

Unfortunately, there was no discussion between reviewers and authors but, in my opinion, the authors answered well to the raised questions and concerns and updated their paper accordingly. All reviewers recommended weak acceptance of the paper.

Strengths:
* Addresses an important problem with a clear motivation.
* Develops an effective approach for F1-score optimization with some new theoretical insights.
* Good empirical performance.
* Mainly clear and easy-to-follow presentation.

Weaknesses:
* Unclear whether F1-score optimization is really necessary for the considered type of problem.
* Some unclarities in the original submission which were mainly fixed.

Overall I think the paper makes some interesting contributions and will be of interest to the community. The weaknesses are minor and can be addressed in an extended discussion in the paper. Hence I am recommending acceptance of the paper.

I would also ask the authors to consider whether they would want to change their paper's title as also suggested by reviewer Ws9d. The term "diagnostic policy learning" is not frequently used.

**Note From Pc:**

if the above contains the word "oral" or "spotlight" please see: "oral" presentation means -> notable-top-5% and "spotlight" means -> notable-top-25%. As stated in our emails, we are disassociating presentation type from AC recommendations